# Cumulative Reasoning With Large Language Models

## Abstract

While language models are powerful and versatile, they often fail to address highly complex problems. This is because solving complex problems requires deliberate thinking, which has been only minimally guided during training. In this paper, we propose a new method called Cumulative Reasoning (CR), which employs language models in a cumulative and iterative manner to emulate human thought processes. By decomposing tasks into smaller components, CR streamlines the problem-solving process, rendering it both more manageable and effective. For logical inference tasks, CR consistently outperforms existing methods with an improvement up to 9.3%, and achieves an accuracy of 98.04% on the curated FOLIO wiki dataset. In the context of the Game of 24, CR achieves an accuracy of 98%, which signifies a substantial enhancement of 24% over the previous state-of-the-art method. Finally, on the MATH dataset, we establish new state-of-the-art results without any external tools with 58.0% overall accuracy, surpassing the previous best approach by a margin of 4.2%, and achieving 43% relative improvement on the hardest level 5 problems ($22.4\% \rightarrow 32.1\%$). Furthermore, we extend the concept of Cumulative Reasoning to include a code environment, in this setup, we are devoid of external aids such as retrieval and web browsing, and focus solely on the LLM's intrinsic computational and logical reasoning capabilities within a Python code environment. Our experiments in this setting yielded impressive results, with an overall accuracy of 72.2% on the MATH dataset, significantly outperforming the PAL method with 38.8% relative improvement[†].

## 1 Introduction

Despite the remarkable advances made by large language models (LLMs) in a variety of applications (Devlin et al., 2018; Radford et al., 2018; 2019; Brown et al., 2020; Raffel et al., 2020; OpenAI, 2023), they still struggle to provide stable and accurate answers when faced with highly complex tasks. For instance, it has been observed that language models have difficulty directly generating correct answers for high school math problems (Lightman et al., 2023).

This shortfall may be anticipated, considering the training approach adopted by LLMs. Specifically, they are trained to sequentially predict the next token based on the given context, without a pause for deliberate thoughts. As elucidated by Kahneman (2011), our cognitive processing processes comprise two distinct systems: System 1 is fast, instinctive, and emotional; System 2 is slow, deliberate, and logical. Currently, LLMs align more closely with System 1, thereby potentially explaining their limitations in confronting complex tasks.

In response to these limitations, several methods have been proposed to mimic human cognitive processes. These include the Chain-of-Thought (CoT) that prompts the model to offer step-by-step solutions (Wei et al., 2022), and the Tree-of-Thought (ToT) that models the solving process as a thought search tree (Yao et al., 2023; Long, 2023). In addition, dedicated datasets have been created to provide step-wise guidance in model training (Lightman et al., 2023). Nevertheless, these methods do not have a site for storing intermediate results, assuming that all the thoughts form a chain or a tree, which does not fully capture the human thinking process.

---

[†]The code is available at `https://anonymous.4open.science/r/cumulative-reasoning-anonymous-4477`.

In this paper, we propose a new method termed Cumulative Reasoning (CR), which presents a more general characterization of the thinking process. CR employs three distinct LLMs: the proposer, verifier, and reporter. The proposer keeps proposing potential propositions, which were verified by one or more verifiers, and the reporter decides when to stop and report the solution.

CR significantly amplifies the power of language models in addressing complex tasks, achieved by decomposing each task into atomic and manageable steps. Despite the computational infeasibility of enumerating the exponentially numerous possible complex tasks, CR ensures that each individual step can be efficiently learned and resolved. This strategic decomposition effectively transforms an otherwise unmanageable exponential problem into a sequence of solvable tasks, thereby providing a robust solution to the original problem.

Our empirical analyses include three components. In the first experiment, we tackled logical inference tasks like FOLIO wiki (pertaining to first-order logic) and AutoTNLI (associated with higher-order logic). On these datasets, CR consistently surpassed current methodologies, showcasing an enhancement of up to 9.3%. Additionally, a rigorous refinement of the FOLIO dataset generated the "FOLIO wiki curated," on which CR recorded a remarkable accuracy of 98.04%. In the second experiment, which revolved around the Game of 24, CR achieved an accuracy of 98%. Remarkably, this represents a significant improvement of 24% when compared to the prior state-of-the-art method, ToT (Yao et al., 2023). In the last experiment, we established new state-of-the-art results on the renowned MATH dataset (Hendrycks et al., 2021), achieving 58.0% overall accuracy with a margin of 4.2% over the Complex-CoT with PHP method (Fu et al., 2022; Zheng et al., 2023). Noteworthy, our method achieves 43% relative improvement on the hardest level 5 problems (22.4% → 32.1%).

Furthermore, we extend the concept of Cumulative Reasoning (CR) with a code environment. Our experimental setup, devoid of other external aids such as external memory, web browsing, or retrieval systems, evaluates the LLM's intrinsic computational and logical reasoning capabilities. We achieved a 72.2% accuracy on the MATH dataset, significantly outperforming methods like PAL (Gao et al., 2023) (52%) and ToRA (Gou et al., 2023) (60.8%). Notably, there was a 66.8% relative improvement over PAL and 12.8% over ToRA on the most challenging level 5 MATH problems, demonstrating the effectiveness of CR in a code environment and further validating the robustness of CR in handling complex tasks.

## 2 PRELIMINARIES

### 2.1 LOGIC

Propositional logic, the most fundamental system of logic, encompasses elements $p, q, r$ and a variety of operations. These include "and" ($p \wedge q$), "or" ($p \vee q$), "implies" ($p \Rightarrow q$), and "not" ($\neg p$). The constants true and false are denoted as $1$ and $0$ respectively. This system adheres to the following rules:

$$x \wedge x = x, \quad x \vee x = x, \quad 1 \wedge x = x, \quad 0 \vee x = x, \quad x \wedge (y \vee x) = x = (x \wedge y) \vee x.$$

and distributive laws:

$$x \wedge (y \vee z) = (x \wedge y) \vee (x \wedge z), \quad x \vee (y \wedge z) = (x \vee y) \wedge (x \vee z).$$

In a Boolean algebra, every element $x$ has a complement $\neg x$ and the following holds true:

$$x \wedge \neg x = 0, \quad x \vee \neg x = 1, \quad \neg\neg x = x.$$

Building upon propositional logic, first-order logic (FOL) introduces universal quantification ($\forall$) and existential quantification ($\exists$) to describe more intricate propositions. For instance, the statement "$\forall_x \text{Dog}(x) \Rightarrow \text{Animal}(x)$" translates to "for every $x$, if $x$ is a dog, then it is also an animal". Higher-order logic (HOL) represents a sophisticated formalism that permits quantification over functions and predicates, an ability that contrasts sharply with FOL, which restricts quantification to individual objects. For a detailed discussion on the distinctive characteristics of HOL, as opposed to FOL, please refer to Appendix D.1.

### 2.2 ILLUSTRATIVE EXAMPLE

Consider the following example adapted from the FOLIO dataset (Han et al., 2022), where empirically only the text statements (excluding logical propositions) will be given:

1. All monkeys are mammals: $\forall x(\text{Monkey}(x) \Rightarrow \text{Mammals}(x))$.
2. An animal is either a monkey or a bird: $\forall x(\text{Animal}(x) \Rightarrow (\text{Monkey}(x) \vee \text{Bird}(x)))$.
3. All birds fly: $\forall x(\text{Bird}(x) \Rightarrow \text{Fly}(x))$.
4. If something can fly, then it has wings: $\forall x(\text{Fly}(x) \Rightarrow \text{Wings}(x))$.
5. Rock is not a mammal, but Rock is an animal: $\neg\text{Mammal}(\text{Rock}) \wedge \text{Animal}(\text{Rock})$.

The question is: Does rock have wings? We have the following derivations:

a. The contrapositive of (1) is: $\forall x(\neg\text{Mammals}(x) \Rightarrow \neg\text{Monkey}(x))$.
b. (a) and (5) $\Rightarrow \neg\text{Monkey}(\text{Rock}) \wedge \text{Animal}(\text{Rock})$.
c. (2) and (5) $\Rightarrow (\text{Monkey}(\text{Rock}) \vee \text{Bird}(\text{Rock}))$
d. (b) and (c) $\Rightarrow \text{Bird}(\text{Rock})$.
e. (3) and (d) $\Rightarrow \text{Fly}(\text{Rock})$.
f. (4) and (e) $\Rightarrow \text{Wings}(\text{Rock})$.

While the derivation can be treated as a general "chain of thought" from $(a)$ to $(f)$, its internal structure is neither a chain nor a tree. Instead, it is a directed acyclic graph (DAG), with each directed edge as one step of derivation. For examples of higher-order logic, see Appendix D.1.

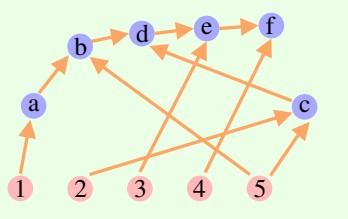

Figure 1: Illustration of our logical derivation

## 3 OUR METHOD

### 3.1 CUMULATIVE REASONING (CR)

Our CR algorithm uses three distinct types of LLMs (AI Agents):

1. Proposer. This model suggests the next step based on the current context.
2. Verifier(s). This model or set of models scrutinizes the accuracy of the step put forward by the proposer. If the step is deemed correct, it will be added to the context.
3. Reporter. This model determines when the reasoning process should be concluded, by assessing whether the current conditions can directly lead to the final solution.

See Figure 2 for an illustration. In each iteration, the proposer initiates the process by proposing one or a few new claim(s) based on existing predicates. Subsequently, the verifier(s) evaluate the proposal, determining whether the claim(s) can be retained as a new predicate. Finally, the reporter decides if it is the optimal time to cease the thought process and deliver the answer.

Ideally, the proposer should be implemented using a language model pre-trained on the corresponding derivation tasks. Verifier(s) should be capable of translating the derivations to appropriate formal systems and verifying them using symbolic reasoning modules such as a propositional logic solver or a formal math prover, such as AI agents equipped with code environment or symbolic systems. However, for simplicity, one can also use general-purpose foundation models like GPT-4 (OpenAI, 2023), instantiated with different prompts for these roles.

The main theoretical motivation of our method lies in the intuitionistic logic, the philosophy of mathematical constructivism, and the topos theory, which imply that the cumulative process of constructing new propositions is the natural way to perform complex reasoning, especially in the realm of (higher-order) logic and pure mathematics.

The primary empirical contribution of our work lies in the synergistic integration of different LLM roles (Proposer, Verifier, and Reporter) within the Cumulative Reasoning framework. This integration facilitates a more effective accumulation and verification of intermediate results, fostering a deeper and more precise reasoning process. The collaborative interplay among these roles (agents), and the

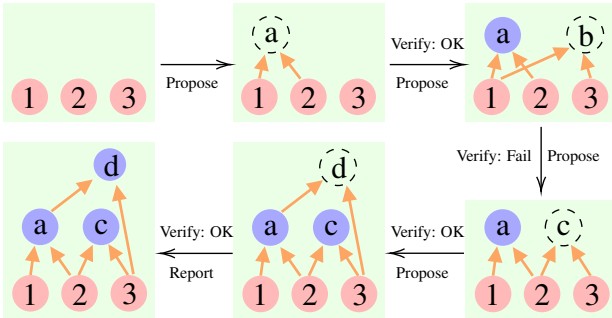

Figure 2: An illustration of Cumulative Reasoning (CR) for a 3-premises problem.

interactions among them and the (code) environments, work together in a synergistic way to enhance the reasoning capabilities of the system. This interplay allows for a more effective accumulation and verification of intermediate results, facilitating a deeper and more precise reasoning process.

### 3.2 COMPARE WITH CoT AND ToT

CR clearly generalizes CoT (Wei et al., 2022), in the sense that if there are no verifiers, and the proposer keeps proposing the next steps until the end, CR becomes the standard chain of thought. However, in CR the overall thinking process is not necessarily a chain or a tree, it can be a DAG. Therefore, CR can be used for solving more complex problems.

At first glance, CR is similar to the ToT, which solves the problems with a thought search tree (Yao et al., 2023; Long, 2023). However, our method is more general in the sense that it stores all the historical correct reasoning results in memory, which can be a DAG (or even directed hyper-graphs). By contrast, ToT will not store the information from other branches for exploration at the current search branch. For a detailed comparison with a preliminary analysis, please refer to Appendix C.

## 4 EXPERIMENTS

Our experimental framework is based on the Microsoft guidance library (Lundberg et al., 2023), which offers the flexibility to intertwine generation, prompting, and logical control in a seamless flow that aligns with language models. We consider the following LLMs: GPT-3.5-turbo, GPT-4, LLaMA-13B and LLaMA-65B.

Our Proposer, Verifier(s), and Reporter in CR are implemented using the same LLM with different few-shot prompts. This approach ensures a broad application scope and simplifies implementation. For optimal results, future work could consider the application of a Proposer pre-trained on task-specific corpus and Verifier(s) aided by symbolic formal systems. We denote $n$ as the number of generated intermediate propositions, and $k$ as the number of majority voting times. We set the temperature $t = 0.1$ by default and $t = 0.7$ for majority voting. We also remark that both GPT-3.5-turbo and GPT-4 operate as chat-format APIs from OpenAI.

### 4.1 FOLIO WIKI

FOLIO dataset (Han et al., 2022) is a first-order logical inference dataset for reasoning in natural language. The label of each problem can be "True", "False", or "Unknown". See Figure 3 for an example. We observed that while the Chain-of-Thought reasoning process can generate useful intermediary results, it tends to flounder midway, failing to arrive at the correct conclusion. Conversely, the CR initially spawns two beneficial propositions and leverages them to successfully solve the problem at hand. For a deeper dive into specific examples of the FOLIO dataset, we refer to Appendix E.1.

The FOLIO dataset is a composite of 1435 examples, wherein 52.5% of these instances have been crafted drawing upon knowledge from randomly selected Wikipedia pages. This approach guarantees the infusion of abundant linguistic variations and a rich vocabulary within the corpus. The residual

47.5% of the examples have been penned in a hybrid style, rooted in a variety of complex logical templates. Acknowledging that contemporary LLMs are pre-trained on a considerable volume of a standard human-written corpus, we direct our experiments towards those examples derived from Wikipedia, hereby referred to as FOLIO-wiki. Once a handful of examples are moved aside for few-shot prompts and those examples without source labels for validations are excluded, we are left with a testable collection of 534 examples.

Our experimental design employs the LLaMA base model and GPT APIs directly, circumventing the need for fine-tuning with logical inference datasets and thus ensuring a faithful comparison. The results, displayed in Table 1, reveal that CR consistently surpasses Direct (standard Input-Output prompt), CoT, and CoT-SC, with a performance margin spanning up to 8.42%. Notably, GPT-4 paired with Cumulative Reasoning (CR) achieves an accuracy rate of 87.45%, outperforming GPT-4 with CoT-SC, which reports an accuracy rate of 85.02%. For more experiments on LogiQA (Liu et al., 2020), ProofWriter (Tafjord et al., 2020), and LogicalDeduction datasets (Srivastava et al., 2022) and more ablation studies, please refer to Appendix B.

## 4.2 FOLIO WIKI CURATED

The accuracy of 87.45% does not seem to be as competitive as human beings, so we carefully reviewed the FOLIO-wiki dataset. It turns out that many instances inside the dataset are problematic in the following sense:

1. Missing common knowledge or contradictory to common knowledge; (9 in total, Example ID No. 34, 62, 162, 167, 228, 268, 526, 677, 679)
2. Overly ambiguous problems failing to provide unequivocal answers; (37 in total, Example ID No. 141, 215, 216, 223, 252, 261, 298, 321, 330, 396, 402, 409, 411, 431, 432, 456, 457, 482, 483, 496, 563, 572, 599, 624, 629, 641, 654, 660, 673, 682, 698, 750)
3. Inherent inconsistencies presented within the premises; (2 in total, Example ID No. 640, 643)
4. Vague premises or typographical errors; (2 in total, Example ID No. 314, 315)
5. Incorrect answers. (24 in total, Example ID No. 9, 46, 52, 84, 100, 144, 273, 276, 299, 310, 322, 345, 367, 437, 452, 453, 464, 557, 573, 578, 605, 632, 671, 715)

We note that except for the first class, all the rest should be removed from the dataset. The first class is because foundation models were trained with common knowledge, but the problem answer based on FOL systems gives an unnatural answer. See Example ID No. 679 shown in Figure 4 and more examples in Appendix E.2) for illustrations. For a brief discussion on the limitations of FOL systems, please refer to Appendix D.

Therefore, we removed all 74 such problematic instances, leaving the remaining 460 examples as a curated collection. The results in Table 2 indicate that the application of GPT-4 in conjunction with our method (CR) commands an astounding accuracy of 98.04% and maintains an error rate as minimal as 1.96%. This level of performance is almost twice as effective compared to the combination of GPT-4 and CoT-SC, which scored an accuracy of 96.09% and an error rate of 3.91%.

## 4.3 AUTOTNLI

**Experiment Setting.** AutoTNLI (Kumar et al., 2022) is a Tabular Natural Language Inference (TNLI) dataset extended from IN-FOTABS (Gupta et al., 2020), which can be seen as a higher-order logical inference dataset due to its inherent complexity lies in natural language inference formalism. It contains 1,478,662 table-hypothesis pairs with the corresponding label (Entail or Neutral) that indicates whether the given table entails the hypothesis. We treat the tabular content within AutoTNLI as a set of premises (In fact, the tables within the AutoTNLI dataset are exactly provided in the form of premises), enabling a direct transference of our method applied to the FOLIO dataset. Our experi-

Table 3: Results for various reasoning approaches on AutoTNLI dataset.

| Model | Method | Acc. ↑ (%) |
|---|---|---|
| - | [Random] | 50.00 |
| LLaMA-13B | Direct | 52.6 |
| | CoT | 54.1 (+1.5) |
| | CoT-SC (k = 16) | 52.1 (-0.5) |
| | **CR (ours**, $n = 4$) | **57.0 (+5.4)** |
| LLaMA-65B | Direct | 59.7 |
| | CoT | 63.2 (+3.5) |
| | CoT-SC (k = 16) | 61.7 (+2.0) |
| | **CR (ours**, $n = 4$) | **72.5 (+12.8)** |

Table 1: Results for various reasoning approaches on FOLIO-wiki dataset.

| Model | Method | Acc. ↑ (%) |
|---|---|---|
| - | [Random] | 33.33 |
| LLaMA-13B | Direct | 44.75 |
| | CoT | 49.06 (+4.31) |
| | CoT-SC ($k = 16$) | 52.43 (+7.68) |
| | **CR (ours**, $n = 2$) | **53.37 (+8.62)** |
| LLaMA-65B | Direct | 67.42 |
| | CoT | 67.42 (+0.00) |
| | CoT-SC ($k = 16$) | 70.79 (+3.37) |
| | **CR (ours**, $n = 2$) | **72.10 (+4.68)** |
| GPT-3.5-turbo | Direct | 62.92 |
| | CoT | 64.61 (+1.69) |
| | CoT-SC (k = 16) | 63.33 (+0.41) |
| | **CR (ours**, $n = 2$) | **73.03 (+10.11)** |
| GPT-4 | Direct | 80.52 |
| | CoT | 84.46 (+3.94) |
| | CoT-SC ($k = 16$) | 85.02 (+4.50) |
| | **CR (ours**, $n = 2$) | **87.45 (+6.93)** |

Table 2: Results for various reasoning approaches on FOLIO-wiki-curated dataset.

| Model | Method | Acc. ↑ (%) |
|---|---|---|
| - | [Random] | 33.33 |
| LLaMA-13B | Direct | 49.13 |
| | CoT | 52.17 (+3.04) |
| | CoT-SC ($k = 16$) | 53.70 (+4.57) |
| | **CR (ours**, $n = 2$) | **55.87 (+6.74)** |
| LLaMA-65B | Direct | 74.78 |
| | CoT | 74.13 (-0.65) |
| | CoT-SC ($k = 16$) | 79.13 (+4.35) |
| | **CR (ours**, $n = 2$) | **79.57 (+4.79)** |
| GPT-3.5-turbo | Direct | 69.57 |
| | CoT | 70.65 (+1.08) |
| | CoT-SC (k = 16) | 69.32 (-0.25) |
| | **CR (ours**, $n = 2$) | **78.70 (+9.13)** |
| GPT-4 | Direct | 89.57 |
| | CoT | 95.00 (+5.43) |
| | CoT-SC (k = 16) | 96.09 (+6.52) |
| | **CR (ours**, $n = 2$) | **98.04 (+8.47)** |

mentation encompassed two models, LLaMA-13B, and LLaMA-65B, each subjected to assessment using Direct, CoT, CoT-SC, and CR methods. Due to the extensive magnitude of the AutoTNLI dataset, we only take the first 1000 table-hypothesis pairs for evaluation.

**Evaluation Results.** As shown in Table 3, both LLaMA-13B and LLaMA-65B models reveal that CR delivers a significant enhancement in performance compared to CoT, with a relative improvement reaching up to 9.3% on the LLaMA-65B model. This data emphasizes the clear advantage of CR over CoT and CoT-SC techniques in the framework of the AutoTNLI dataset.

## 4.4 GAME OF 24

The Game of 24 is a puzzle in which players must combine four specified integers using basic arithmetic operations (addition, subtraction, multiplication, division) to get the number 24.

**Settings and Baselines.** To ensure fairness, we adopt exactly identical task settings as Tree of Thoughts (ToT) (Yao et al., 2023) on Game of 24. We use the set of 100 Games of 24 collected by Yao et al. (2023) which was been used to evaluate the performance of ToT. In each game, we consider the game to be successfully solved if and only if the output is a valid equation that reaches 24 and only uses given numbers each exactly once. We quantify the accuracy (success rate) across 100 games as a main evaluative metric.

In this experiment, we compare CR with variant prompt algorithms, including standard Input-Output prompting (Direct), Chain-of-Thought prompting (CoT), and CoT-SC by aggregating the majority outcome from 100 sampled CoT trials (designated as k = 100), and Tree of Thoughts (ToT) with a breadth-first search width set at 5 (indicated as b = 5).

**CR Setup.** Within our CR algorithm, we maintain a set of "reached states", denoted by $S$. Initially, $S$ only contains the start state $s$ which represents 4 input numbers without any operation. In each iteration, a state $u$ is randomly selected from $S$. This selected state $u$ is passed to the Proposer, which randomly picks two remaining numbers within $u$ and combines them through a basic arithmetic operation (+,-,*, /) to obtain a new number, thereby generating a new state $v$. The Proposer is instructed to try to avoid taking duplicated operations. Subsequently, the Verifier scrutinizes the arithmetic operation proposed by the Proposer and evaluates the newly generated state $v$. Then $v$ is inserted to $S$ if the Verifier thinks that the operation from $u$ to $v$ is legitimate and it is potential for $v$ to achieve 24. Upon the Verifier identifying a state $t$ that unequivocally 24, the Reporter devises a solution based on the path from the state $s$ to state $t$ and produces the final answer. The algorithm terminates when the Reporter outputs the final answer or the number of iterations exceeds a limit of $L$. In the experiments, we set the default value of $L$ to 50.

Following Yao et al. (2023), our algorithm runs $b$ concurrent branches and only selects the first answer for these branches that utilizes each input number exactly once for evaluation. Due to the prohibitive cost of GPT-4, we only test our CR algorithm with $b = 1$ to $b = 5$. As shown in Table 4, we find that CR outperforms ToT by a large margin of 24%, from 74% to 98%, with much fewer states visited.

**Compare with ToT.** Interestingly, in the context of Game of 24, our CR algorithm and ToT algorithm are very similar. Their primary distinction is that, in CR, each iteration of the algorithm generates at most one newly reached state, while ToT produces a multitude of candidate states per iteration, filtering and retaining a subset of states. This implies that ToT explores a larger number of invalid states compared to CR. Moreover, ToT employs a fixed-width and fixed-depth search tree, while CR allows the LLM to determine the search depth autonomously, and performs different search widths on different layers of the search tree.

Table 4: Results for various approaches on Game of 24 using GPT-4. The average number of visited states for ToT is computed from the experimental logs available in its official GitHub repository.

| Method | Acc. ↑ (%) | # Visited states ↓ |
|---|---|---|
| Direct | 7.3 | 1 |
| CoT | 4.0 | 1 |
| CoT-SC (k = 100) | 9.0 | 100 |
| Direct (best of 100) | 33 | 100 |
| CoT (best of 100) | 49 | 100 |
| ToT ($b = 5$) | 74 | 61.72 |
| **CR (ours**, $b = 1$) | 84 (+10) | **11.68 (-50.04)** |
| **CR (ours**, $b = 2$) | 94 (+20) | 13.70 (-48.02) |
| **CR (ours**, $b = 3$) | 97 (+23) | 14.25 (-47.47) |
| **CR (ours**, $b = 4$) | 97 (+23) | 14.77 (-46.95) |
| **CR (ours**, $b = 5$) | **98 (+24)** | 14.86 (-46.86) |

## 5 SOLVING MATH PROBLEMS

### 5.1 CR WITHOUT CODE ENVIRONMENT

The MATH dataset (Hendrycks et al., 2021) serves as a benchmark for assessing AI models' mathematical reasoning capabilities, encompassing a broad spectrum of mathematical problems across various subdomains such as Algebra and Geometry. Figure 5 in Appendix A shows an illustrative example from the MATH dataset, and Figure 6 in Appendix A shows the corresponding solutions generated by Complex CoT and CR. In our experiments, we assessed the performance of Complex CoT and our method (CR), both with and without Progressive-Hint Prompting (PHP) (Zheng et al., 2023). For a fair evaluation, we reproduced the results of Complex CoT (w/ PHP) on a subset of 500 test examples, adhering to Lightman et al. (2023), since the other parts of the test dataset (4500 examples) may have been utilized for model training by OpenAI. The difficulty spans from level 1 (simplest) to level 5 (hardest).

It is important to note that for our method (CR), we employed 4-shot prompting (4 examples for few-shot prompting) due to GPT-4's context length constraints (8k by default). While the model occasionally exceeds the context length with 8-shot prompting, it generally demonstrates superior performance. Future experiments will explore the utilization of GPT-4-32k.

From Table 5, our method (CR) distinguishes itself by achieving significant advancements in performance across various mathematical subdomains, outperforming Complex CoT by a margin of 5.4%. The enhancements are particularly pronounced in the Number Theory, Probability, PreAlgebra, and Algebra categories. In comparison to the Complex CoT approach, even when restricted to 4-shot prompting due to GPT-4's context length constraints, CR demonstrates its robustness and effectiveness. It is also evident that the PHP method further amplifies the performance of both Complex CoT and CR, establishing new state-of-the-art results with an overall accuracy of 58.0% using CR with PHP, with a margin of 4.2% over Complex CoT with PHP. Additionally, the "Iters" metric elucidates that CR, when synergized with PHP strategies, reaches self-consistent answers with fewer iterations.

From Table 6, it is evident that consistent performance boost across different difficulty levels signifies the robustness of the CR methodology in handling a diverse range of mathematical problems. The performance increase of 9.7% at level 5—which translates to a substantial relative improvement of 43%—compared to the baseline Complex CoT approach without PHP, underscores CR's effectiveness in handling the most challenging problems in the dataset.

Table 5: Comparative performance on the MATH dataset using GPT-4 without code environment. We adopted a default temperature setting of $t = 0.0$, consistent with prior research settings (greedy decoding). PHP denotes the application of the progressive-hint prompting. "Iters" represents the average number of LLM interactions, and **Overall** reflects the overall results across MATH subtopics.

| | w/ PHP | MATH Dataset (* denotes using 500 test examples subset following Lightman et al. (2023)) | | | | | | | |
| --- | --- | --- | --- | --- | --- | --- | --- | --- | --- |
| | | InterAlgebra | Precalculus | Geometry | NumTheory | Probability | PreAlgebra | Algebra | **Overall** |
| CoT (OpenAI, 2023) | ✗ | - | - | - | - | - | - | - | 42.50 |
| Complex CoT, 8-shot | ✗ | 23.4 | 26.7 | 36.5 | 49.6 | 53.1 | 71.6 | 70.8 | 50.36 |
| (Zheng et al., 2023) | ✓ | 26.3 | 29.8 | 41.9 | 55.7 | 56.3 | 73.8 | 74.3 | 53.90 |
| | (Iters) | 3.2414 | 3.2435 | 3.2233 | 3.1740 | 2.8122 | 2.3226 | 2.4726 | 2.8494 |
| Complex CoT* | ✗ | 29.9 | 33.9 | 34.1 | 46.8 | 47.4 | 62.1 | 70.7 | 48.80 |
| (repro., 8-shot) | ✓ | 28.9 | 30.4 | 43.9 | 53.2 | 50.0 | 68.5 | 84.1 | 53.80 |
| | (Iters) | 2.7629 | 2.4643 | 2.7805 | 2.7581 | 2.4474 | 2.3780 | 2.5484 | 2.59 |
| **CR w/o code*** | ✗ | 28.9 (-1.0) | 30.4 (-3.5) | 39.0 (+4.9) | 54.8 (+8.0) | 57.9 (+10.5) | 71.8 (+9.7) | 79.3 (+8.6) | **54.20 (+5.40)** |
| (**ours**, 4-shot) | ✓ | **32.0 (+3.1)** | **35.7 (+5.3)** | **43.9 (+0.0)** | **59.7 (+6.5)** | **63.2 (+13.2)** | **71.8 (+3.3)** | **86.6 (+2.5)** | **58.00 (+4.20)** |
| | (Iters) | 2.6598 | 2.4821 | 2.5122 | 2.2903 | 2.2105 | 2.2195 | 2.3548 | **2.40 (-0.19)** |

Table 6: Comparative performance on the MATH dataset using GPT-4 without code environment for different difficulty levels.

| | w/ PHP | MATH Dataset (* denotes using 500 test examples subset) | | | | | |
| --- | --- | --- | --- | --- | --- | --- | --- |
| | | Level 5 | Level 4 | Level 3 | Level 2 | Level 1 | **Overall** |
| CoT (OpenAI, 2023) | ✗ | - | - | - | - | - | 42.50 |
| Complex CoT* | ✗ | 22.4 | 38.3 | 62.9 | 72.2 | 79.1 | 48.80 |
| (repro., 8-shot) | ✓ | 23.9 | 43.8 | 63.8 | 86.7 | 83.7 | 53.80 |
| **CR w/o code*** | ✗ | **32.1 (+9.7)** | 43.0 (+4.7) | 62.9 (+0.0) | 78.9 (+6.7) | 83.7 (+4.6) | **54.20 (+5.40)** |
| (**ours**, 4-shot) | ✓ | 27.3 (+3.4) | **50.0 (+6.2)** | **70.9 (+7.1)** | **86.7 (+0.0)** | **90.7 (+7.0)** | **58.00 (+4.20)** |

## 5.2 CR WITH CODE ENVIRONMENT ONLY

In this section, we extend the concept of Cumulative Reasoning (CR) with the inclusion of a code environment. Our experimental setup chooses not to utilize external aids such as memory modules, web browsing, or retrieval systems. Instead, we focus on a pure Python code environment to emulate a symbolic system. This approach aims to evaluate the LLM's intrinsic capabilities in computational problem-solving and logical reasoning. This involves a single reasoning context session without additional verifier LLMs.

In the CR framework with a code environment, the Python interpreter acts as a symbolic system that aids in verification. This setup allows for an intricate interplay between the proposer (LLM) and the verifier (LLM equipped with code environment). The LLM, acting as the proposer, can generate hypotheses, formulate mathematical expressions, and pose questions to itself. These steps are then executed and verified in the code environment, and the observations (outputs) are then interpreted by the LLM.

Our experimental results, as shown in Table 7 and Table 8, demonstrate the effectiveness of the CR methodology in a code environment. We compare our approach with PAL (Gao et al., 2023) and ToRA (Gou et al., 2023), two notable benchmarks in the field. CR with code significantly outperforms these methods, achieving an overall accuracy of 72.2% on the MATH dataset, achieving 38.9% relative improvement over PAL and 18.8% relative improvement over ToRA. More specifically, achieving 66.8% relative improvement of PAL, and 12.8% relative improvement over ToRA on the hardest level 5 MATH problems.

## 6 RELATED WORK

**Reasoning with LLM.** An extensive range of studies highlights the benefits of equipping neural networks with the capacity to generate intermediate steps, which is a capability that notably enhances reasoning performance across a broad spectrum of applications (Zaidan et al., 2007; Yao et al., 2021; Hase & Bansal, 2021; Yang et al., 2022; Wu et al., 2022; Zhou et al., 2022). Morishita et al. (2023) improve the reasoning abilities of language models by using a synthetic corpus derived from formal logic theory. A comprehensive analysis of process-based versus outcome-based approaches on the GSM8K task is conducted by Uesato et al. (2022), and Lightman et al. (2023) further advance this field by meticulously collecting the PRM-800K dataset containing step-by-step supervision.

Table 7: Comparative performance on the MATH dataset using GPT-4 and GPT-4-turbo with Python code environment. We adopted a default temperature setting of $t = 0.0$, consistent with prior research settings (greedy decoding). Notice that in this experiment (including reproduced results), we use a lightweight GPT-4-turbo for a cheaper cost as default. "Sessions" denotes how many LLMs with a consecutive thinking context are involved in the reasoning process, and **Overall** reflects the overall results across MATH subtopics.

| | # Sessions | MATH Dataset (* denotes 500 text examples subset) | | | | | | | |
| --- | --- | --- | --- | --- | --- | --- | --- | --- | --- |
| | | InterAlgebra | Precalculus | Geometry | NumTheory | Probability | PreAlgebra | Algebra | **Overall** |
| PAL | - | 32.8 | 29.3 | 38.0 | 58.7 | 61.0 | 73.9 | 59.1 | 51.8 |
| PAL* (repro., 4 shot) | 1 | 30.9 | 23.2 | 31.7 | 66.1 | 57.9 | 73.2 | 65.3 | 52.0 |
| ToRA | - | 40.0 | 37.2 | 44.1 | 68.9 | 67.3 | 82.2 | 75.8 | 61.6 |
| ToRA* (repro., 4 shot) | 1 | 49.5 | 44.6 | 48.8 | 49.5 | 66.1 | 67.1 | 71.8 | 60.8 |
| **CR w/ code*** (**ours**, 4-shot) | 1 | **51.5** (+2.0) | **51.8** (+7.2) | **53.7** (+4.9) | **88.7** (+22.6) | **71.1** (+5.0) | **86.6** (+13.4) | **86.3** (+14.5) | **72.2** (+11.4) |

Table 8: Comparative performance on the MATH dataset using GPT-4 and GPT-4-turbo with Python code environment for different difficulty levels.

| | # Sessions | MATH Dataset (* denotes using 500 test examples subset) | | | | | |
| --- | --- | --- | --- | --- | --- | --- | --- |
| | | Level 5 | Level 4 | Level 3 | Level 2 | Level 1 | **Overall** |
| PAL | - | - | - | - | - | - | 51.8 |
| PAL* (repro., 4-shot) | 1 | 31.3 | 45.3 | 60.0 | 65.6 | 88.4 | 52.0 |
| ToRA | - | - | - | - | - | - | 61.6 |
| ToRA* (repro., 4-shot) | 1 | 46.3 | 53.9 | 69.5 | 75.6 | 74.4 | 60.8 |
| **CR w/ code*** (**ours**, 2-shot) | 1 | **52.2** (+5.9) | **66.4** (+12.5) | **81.9** (+12.4) | **90.0** (+14.4) | **90.7** (+2.3) | **72.2** (+11.4) |

Additionally, a considerable breadth of research is committed to amplifying the reasoning capabilities leveraging symbolic systems, including code environment, knowledge graphs, and formal theorem provers (Mihaylov & Frank, 2018; Bauer et al., 2018; Kundu et al., 2018; Wang et al., 2019; Lin et al., 2019; Ding et al., 2019; Feng et al., 2020; Wang et al., 2022a; Chen et al., 2022; Lyu et al., 2023; Chen et al., 2022; Gao et al., 2023; Gou et al., 2023; Jiang et al., 2022; Yang et al., 2023).

**Chain-of-Thought Prompting.** In the pioneering work on chain-of-thought reasoning, Wei et al. (2022) emphasize the importance of incorporating multi-step reasoning paths before generating definitive answers. In a progression from this, Wang et al. (2022b) introduce self-consistency, a sophisticated decoding strategy destined to supersede the rudimentary greedy decoding employed in CoT prompting. Advancing this further, Zhou et al. (2022) seek to tackle the complexities faced by CoT prompting in addressing tasks necessitating solutions beyond the complexity scope of the exemplars used in the prompts. Khot et al. (2022) enhance LLM capabilities for complex tasks through Decomposed Prompting, a method that dissects tasks into simpler sub-tasks. Creswell & Shanahan (2022) showcase a method for enhancing reasoning quality, conducting a beam search throughout the reasoning trace space. Fu et al. (2022) highlight the importance of increasing reasoning complexity inside the few-shot prompts for better performance.

More recently, Li et al. (2023) bring forth DIVERSE, which generates a spectrum of prompts to scrutinize various reasoning trajectories for an identical question, and utilizes a verifier to weed out incorrect answers using a weighted voting scheme. Yao et al. (2023) propose a framework for language model inference, Tree-of-Thought (ToT). ToT enhances the problem-solving abilities of language models by facilitating deliberate decision-making, contemplating multiple reasoning paths, and performing self-evaluative choices to determine subsequent actions. Taking an iterative approach, Zheng et al. (2023) advocate for recurrent invocations of LLMs, leveraging prior answers as contextual hints to inform subsequent iterations. Lastly, Feng et al. (2023) underscore the theoretical prowess of CoT in addressing intricate real-world tasks like dynamic programming.

# 7 CONCLUSION

In this paper, we propose CR that employs language models iteratively and cumulatively. The main idea behind our algorithm is decomposing the complex task into smaller steps, and maintaining a thinking context for all the intermediate results. Experimental results show that our method achieves state-of-the-art performance for logical inference tasks, the Game of 24, and MATH problems. Given its inherent generality, our framework holds promising potential for addressing a wider array of mathematical challenges.

## ETHICS STATEMENT

Our research on Cumulative Reasoning (CR) aims to enhance the problem-solving abilities of language models and shows significant improvements in tasks such as logical inference and complex problem-solving. We use a curated FOLIO wiki dataset derived from Yale's publicly available FOLIO dataset, ensuring that all data is anonymized and stripped of personally identifiable information. While CR potentially makes the decision-making process more transparent by breaking down tasks into simpler components, it inherits the biases present in the language models' training data and maintains some level of the 'black box' nature. Its advanced reasoning capabilities, although promising for beneficial applications like medical diagnostics, also pose risks of misuse, such as in disinformation campaigns. Furthermore, the computational intensity of training these models has environmental implications. We urge the research community to adopt responsible guidelines for the deployment of advanced reasoning models and consider future work in improving interpretability, mitigating biases, and reducing environmental impact.

## REPRODUCIBILITY STATEMENT

To facilitate reproducibility, we make our code available at `https://anonymous.4open.scie nce/r/cumulative-reasoning-anonymous-4477`. The experiment results can be easily reproduced following the instructions in the README document. We also depict our experiment details in Section 4.

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

# A    APPENDIX FOR EXAMPLES

## A.1    ILLUSTRATIVE EXAMPLE ON EXPERIMENT DATASETS

---

**[Problem Description]**

- Example ID: 687
- **Premises**:
    1. It is not true that some giant language models do not have good performance.
    2. All language models with good performance are used by some researchers.
    3. If a language model is used by some researchers, it is popular.
    4. If BERT is a giant language model, then GPT-3 is also a giant language model.
    5. BERT is a giant language model.
- **Hypothesis**: GPT-3 is popular.
- **Label**: **[True]**

---

**[Chain-of-Thought Reasoning by GPT-4]**

- **Reasoning**: "From the premises, we have:

$$1. \neg(\exists x)(\text{Giant}(x) \land \neg\text{GoodPerformance}(x))$$
$$2. \forall x(\text{GoodPerformance}(x) \rightarrow \text{UsedByResearchers}(x))$$
$$3. \forall x(\text{UsedByResearchers}(x) \rightarrow \text{Popular}(x))$$
$$4. \text{BERT} \rightarrow \text{GPT-3}$$
$$5. \text{BERT}$$

Now we know that BERT is a giant language model, so GPT-3 is also a giant language model (from premises 4 and 5). However, the premises do not provide information about GPT-3's performance or if it is used by researchers. Therefore, we cannot conclude if GPT-3 is popular."
- **Prediction**: **[Unknown]** (**Wrong**)

---

**[Cumulative-Reasoning by GPT-4]**

- **Generated Propositions**:
    1. "GPT-3 is a giant language model with good performance."
    2. "GPT-3 is used by some researchers."
- **Reasoning**: "We know that GPT-3 is a giant language model with good performance, and it is used by some researchers. According to the premises, if a language model is used by some researchers, it is popular. Therefore, GPT-3 is popular."
- **Prediction**: **[True]** (**Correct**)

---

Figure 3: An example from the FOLIO dataset, with solutions generated by CoT and CR. CoT will generate the answer directly through a chain of thought. By contrast, CR will first generate a few propositions, and conclude with the help of the generated propositions.

---

**[Problem Description]**

- Example ID: 679
- **Premises**:
    1. Zaha Hadid is a British-Iraqi architect, artist and designer.
    2. Zaha Hadid was born on 31 October 1950 in Baghdad, Iraq.
    3. Hadid was a visiting professor of Architectural Design at the Yale School of Architecture.
    4. Max is an aspiring architecture student, and he plans to apply to Yale School of Architecture.
- **Hypothesis**: Hadid was born in 1982.
- **FOL Label**: [Unknown]
- **Human Label**: [False]
- **Explanation**: *We can see that Zaha Hadid was born on 31 October 1950 in Baghdad, Iraq. This directly contradicts the hypothesis that Hadid was born in 1982. It is common knowledge that people are born only once, and someone can't be born in two different years.*

---

Figure 4: Example 679 from the FOLIO wiki dataset, the origin label provided by the FOL system is not correct, so we choose to curate this dataset, removing these examples with wrong labels. For more examples, please refer to Appendix E.2.

---

**[Problem Description]**

- Example ID: test/intermediate_algebra/1350.json
- Level: 5
- Subject: Intermediate Algebra
- **Problem**: Consider the polynomial

$$f(x) = a_n x^n + a_{n-1} x^{n-1} + \cdots + a_2 x^2 + a_1 x + a_0,$$

where the polynomial has integer coefficients and its roots are distinct integers.
Given $a_n = 2$ and $a_0 = 66$, the inquiry is to determine the least possible value of $|a_{n-1}|$.

---

**[Ground Truth Solution]**

- **Solution**: Since $f(x)$ has integer coefficients, the Integer Root Theorem asserts that any integer roots of $f(x)$ must divide the constant term $66 = 2 \cdot 3 \cdot 11$. Consequently, the potential integer roots of $f(x)$ are

$$\pm 1, \ \pm 2, \ \pm 3, \ \pm 6, \ \pm 11, \ \pm 22, \ \pm 33, \ \pm 66.$$

Additionally, given that all roots of $f(x)$ are integers, they are necessarily members of the aforementioned list.
We proceed to utilize Vieta's formulas. The roots of $f(x)$ yield a product of $(-1)^n \cdot \frac{a_0}{a_n}$, which evaluates to either $33$ or $-33$. Simultaneously, the sum of these roots is $-\frac{a_{n-1}}{a_n} = -\frac{a_{n-1}}{2}$. To minimize $|a_{n-1}|$, we aim to reduce the absolute value of the root sum, ensuring that the product of the roots remains $33$ or $-33$.
Considering two distinct scenarios:
**Case 1:** One of the roots is $33$ or $-33$. In this scenario, the only other viable roots are $\pm 1$. Here, the root sum's absolute value is at least $32$.
**Case 2:** One root is $11$ or $-11$, and another is $3$ or $-3$. The only other plausible roots remain $\pm 1$, leading to a root sum's absolute value of at least $11 - 3 - 1 = 7$. This is a more optimal outcome than Case 1. For an absolute root sum of $7$, we find $|a_{n-1}| = 7|a_n| = 14$.
From our analysis, we deduce that $|a_{n-1}| \geq 14$. Verifying this, we observe that equality is attained by

$$f(x) = 2(x + 11)(x - 3)(x - 1)$$
$$= 2x^3 + 14x^2 - 82x + 66,$$

which is consistent with having integer coefficients and integer roots. Thus, the minimal achievable value for $|a_{n-1}|$ is $\boxed{14}$.
- **Answer**: **14**

---

Figure 5: An example from the MATH dataset (Hendrycks et al., 2021; Lightman et al., 2023).

---

**[Chain-of-Thought Reasoning by GPT-4]**

- **Reasoning**: "Let's think step by step:

  1. The roots of the polynomial are distinct integers, and the leading coefficient is 2, so the roots must be divisors of $\frac{66}{2} = 33$.

  2. The divisors of 33 are $\pm 1, \pm 3, \pm 11, \pm 33$.

  3. The sum of the roots of the polynomial is $-\frac{a_{n-1}}{a_n}$.

  4. To minimize $|a_{n-1}|$, we should select the roots to be as close to zero as possible.

  5. The smallest possible sum of the roots is $-1 - 3 + 11 + 33 = 40$,

  6. Thus, the smallest possible value of $|a_{n-1}|$ is $2 \times 40 = \boxed{80}$."

- **Answer**: **80** (**Wrong**)

---

**[Cumulative-Reasoning by GPT-4]**

- **Generated Hints**:

  1. "**Hint 1**: Recall Vieta's formulas, which relate the coefficients of a polynomial to the sums and products of its roots."

  2. "**Hint 2**: The product of the roots of the polynomial is equal to $-\frac{a_0}{a_n}$."

  3. "**Hint 3**: The sum of the roots of the polynomial is equal to $-\frac{a_{n-1}}{a_n}$."

  4. "**Hint 4**: Since the roots are distinct integers, consider the factors of $-\frac{a_0}{a_n}$ and their sums."

- **Generated Simple Questions and Answers**:

**Question 1:** "What is the product of the roots of the polynomial $f(x)$?"

   **Answer 1:** "The product of the roots of the polynomial is $-\frac{a_0}{a_n} = -\frac{66}{2} = -33$."

**Question 2:** "What are the possible sets of distinct integer roots of the polynomial?"

   **Answer 2:** "The possible sets of distinct integer roots are $\{-33, 1\}$, $\{-11, 3\}$, and $\{-3, 11, -1\}$."

**Question 3:** "What are the sums of the roots for each possible set of roots?"

   **Answer 3:** "The sums of the roots are $-32$, $-8$, and $7$ for the sets $\{-33, 1\}$, $\{-11, 3\}$, and $\{-3, 11, -1\}$, respectively."

**Question 4:** "What is the least possible value of $|a_{n-1}|$?"

   **Answer 4:** "The least possible value of $|a_{n-1}|$ is $2 \times 7 = 14$."

- **Solution**: "Let's think step by step. Given that $a_n = 2$ and $a_0 = 66$, and the roots of the polynomial are distinct integers, the least possible value of $|a_{n-1}|$ is $\boxed{14}$."

- **Answer**: **14** (**Correct**)

---

Figure 6: Solutions for the example presented in Figure 5 from the MATH dataset, generated by CoT and CR. CoT will generate the answer directly through a chain of thought. By contrast, CR will first generate a few hints, then several simple and foundational questions, and then answer them by self, and finally conclude with the help of the generated hints and question-answer pairs.

```
<syntax>

## Problem: [problem]

Solution: Lets' think step by step. [somewords interpreting the origin problem]

### Preliminary Contents

- **Prelim 1**: [preliminary contents 1]

- **Prelim 2**: [preliminary contents 2]

- [...]

### Hints
- **Hint 1**: [useful hints 1]

- **Hint 2**: [useful hints 2]

- [...]

### Intermediate Steps: Question-AnswerSketch-Code-Output-Answer Pairs

Let's think step by step.

#### Question 1: [the first question you raised]
- **Answer Sketch**: [write a sketch of your answer to question 1]

##### Code for Question 1
[call code interpreter here to verify and solve your answer sketch to question 1]

#### Answer for Question 1
- **Answer**: [your answer to this question 1 based on the results
given by code interpreter (if presented)]

#### Question 2: [the second question you raised]
- **Answer Sketch**: [write a sketch of your answer to question 2]

##### Code for Question 2
[call code interpreter here to verify and solve your answer sketch to question 2]

#### Answer for Question 2
- **Answer**: [your answer to this question 2 based on the results
given by code interpreter (if presented)]

#### Question 3: [the second question you raised]
- **Answer Sketch**: [write a sketch of your answer to question 3]

##### Code for Question 3
[call code interpreter here to verify and solve your answer sketch to question 3]

#### Answer for Question 3
- **Answer**: [your answer to this question 3 based on the results
given by code interpreter (if presented)]

### [Question ...]

### Final Solution:

Recall the origin problem <MathP> [origin problem] </MathP>.

Let's think step by step.

#### Solution Sketch
[write a sketch for your final solution]

#### Code for Final Solution
[call code interpreter here to verify and solve your final solution]

#### Final Answer
[present the final answer in latex boxed format, e.g., $\boxed{63\pi}$]
Final Answer: the answer is $\boxed{...}$.

</syntax>
---
```

Figure 7: Meta Prompt for CR with code environment on solving MATH problems.

```
As one of the most distinguished mathematicians, logicians, programmers, and AI scientists, you
possess an unparalleled mastery over Arithmetic, Combinatorics, Number Theory,
Probability Theory, Algebra, Analysis, and Geometry. You are not only intelligent and rational
but also prudent and cautious. You are willing to write and execute Python code. Let's approach
each problem step by step, take a deep breath, do not save your words, and articulate
our thoughts in detail, as detailed as possible.

<system>
You will be presented with a mathematical problem, denoted as `MathP`. Before diving into
the solution, you are asked to lay down some foundational preliminary contents and hints.
Thereafter, you will generate a series of intermediate questions that pave the way to the
final answer of `MathP`. For each question, sketch a preliminary answer, execute the
corresponding code (you always remember to `from sympy import *`), derive the output,
and then finalize your answer.
This forms a [Question] -> [AnswerSketch] -> [Code] -> [Output] -> [Answer] sequence.

## System Instructions for Mathematical Problem-Solving

### Objective
Your primary goal is to solve complex mathematical problems with code environment feedback.

### Key Priorities

1. **Hints**: Prioritize generating hints that are useful for solving the problem.

2. **Intermediate Questions**: Craft questions that decompose the problem into simpler parts,
then try to solve them with code environment feedback.

### Code Execution Guidelines

1. **Import Libraries**: YOU MUST IMPORT NECESSARY LIBRARIES in all your code blocks,
such as `from sympy import *`.

2. **Immediate Execution**: Execute **all** your code immediately after writing them to ensure
they are working as intended.
You should use code interpreter immediately after you have written the code,
to get the output.

3. **YOU MUST CALL CODE INTERPRETER IMMEDIATELY IN EVERY QUESTION**.

### Mathematical Formatting

1. **Final Answer**: Present your final answer to the origin problem lastly (not your generated
questions)
in LaTeX format, enclosed within `\boxed{}` and devoid of any units.

2. **Mathematical Constants and Rational Numbers**: Use the `pi` symbol and the `Rational` class
from the Sympy library to represent \( \pi \) and fractions. All fractions and square roots
should be simplified but **not** converted into decimal values.
</system>

---
```

Figure 8: System Instructions used in CR with code environment for solving MATH problems, the actual context would be [SystemInstruction] + [MetaPrompt].

# B MORE EXPERIMENTS ON LOGICAL INFERENCE TASKS

## B.1 MORE EXPERIMENTAL RESULTS

Table 9: Comparison results on LogiQA

| Method | Acc. ↑ | # Visited States ↓ |
|---|---|---|
| Direct | 31.69% | 1 |
| CoT | 38.55% | 1 |
| CoT-SC | 40.43% | **16** |
| ToT | 43.02% | 19.87 |
| CR | **45.25%** | 17 |

Table 10: Comparison results on ProofWriter

| Method | Acc. ↑ | # Visited States ↓ |
|---|---|---|
| Standard | 46.83% | 1 |
| CoT | 67.41% | 1 |
| CoT-SC | 69.33% | **16** |
| ToT | 70.33% | 24.57 |
| CR | **71.67%** | 16.76 |

Table 11: Comparison results on FOLIO-val

| Method | Acc. ↑ | # Visited States ↓ |
|---|---|---|
| Standard | 60.29% | 1 |
| CoT | 67.65% | 1 |
| CoT-SC | 68.14% | 16 |
| ToT | **69.12%** | 19.12 |
| CR | **69.11%** | **15.87** |

Table 12: Comparison results on LD

| Method | Acc. ↑ | # Visited States ↓ |
|---|---|---|
| Standard | 71.33% | 1 |
| CoT | 73.33% | 1 |
| CoT-SC | 74.67% | **16** |
| ToT | 76.83% | 21.83 |
| CR | **78.33%** | 16.98 |

For a fair comparison of different methods on the LogiQA, ProofWriter, FOLIO (validation set), and LD datasets, we report the third-party reproduced results by Sun et al. (2023), For implementation details on these experiments, please refer to their work.

## B.2 ABLATION STUDIES

Table 13: Ablation studies on FOLIO wiki dataset using GPT-3.5-turbo model.

| Model | Method | Acc. ↑ (%) |
|---|---|---|
| - | [Random] | 33.33 |
| GPT-3.5-turbo | Direct | 62.92 |
| | CoT | 64.61 (+1.69) |
| | CoT-SC (k = 16) | 63.33 (+0.41) |
| | **CR (ours**, $n = 2$) | **73.03 (+10.11)** |
| | **CR (ours**, $n = 2$, w/o Verifier) | 64.23 (+1.31) |
| | **CR (ours**, $n = 2$, w/o premises random choice) | 68.73 (+5.81) |
| | **CR (ours**, $n = 2$, w/o Verifier, w/o premises random choice) | 67.23 (+4.31) |

## C  DETAILED COMPARISON OF CoT, ToT AND CR

To compare these methods, we consider a simple 2-stage reasoning process, which can be extended to multiple stages as well. For simplicity, whenever the model has a step-verifier, we assume that the verifier has 100% accuracy. Moreover, we assume that there exists exactly one correct reasoning path for the problem. We have the following definitions.

**Definition C.1** (Arrival Probability). For a given algorithm, we may compute its arrival probability as the probability of reaching the correct conclusion from the initial state, with one-experience successful invocation. Specifically, denote the arrival probability of CoT as $P_{\text{CoT}}$, the arrival probability of running CoT multiple times as $P_{\text{CoT-SC}}$, the arrival probability of ToT as $P_{\text{ToT}} = p_{1_{\text{ToT}}} p_{2_{\text{ToT}}}$, the arrival probability of CR as $P_{\text{CR}} = p_{1_{\text{CR}}} p_{2_{\text{CR}}}$. Here, $p_{1_{\text{ToT}}}$ and $p_{1_{\text{CR}}}$ are the probablity of getting the first reasoning step correctly, while $p_{2_{\text{ToT}}}$ and $p_{2_{\text{CR}}}$ are for the second step conditioned on the first step being correct.

Since both ToT and CR have verifiers, they can exclude the wrong reasoning path immediately, see Figure 9. Therefore, we immediately have $P_{\text{CoT}} \leq p_{1_{\text{ToT}}} p_{2_{\text{ToT}}}$, as CoT explores more useless branches.

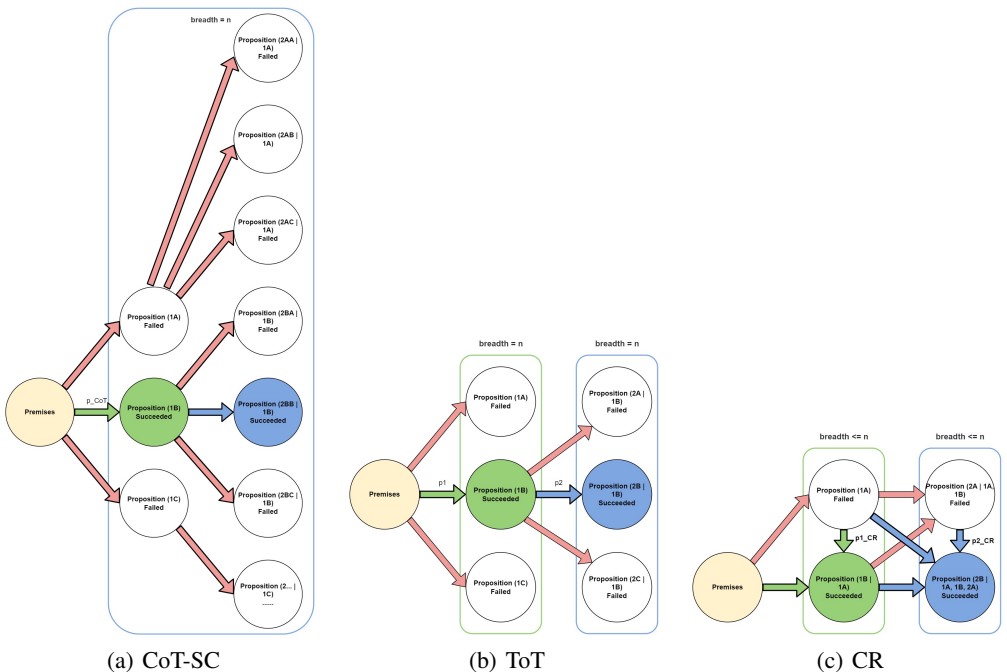

(a) CoT-SC          (b) ToT          (c) CR

Figure 9: Comparison between CoT-SC, ToT, and CR.

Notice that using $p_{1_{\text{CR}}}$ or $p_{2_{\text{CR}}}$ to denote the arrival probabilities of CR is not accurate, as CR will maintain a history of visited states. Therefore we use $p_{1_{\text{CR}}|(\cdot)}$ and $p_{2_{\text{CR}}|(\cdot)}$ to denote the probability conditioned with additional visited states. We have the following assumption.

**Assumption C.2.** $p_{1_{\text{ToT}}} \leq p_{1_{\text{CR}}}$, $p_{2_{\text{ToT}}} \leq p_{2_{\text{CR}}}$, In addition, $p_{1_{\text{CR}}|(\cdot)}$ and $p_{2_{\text{CR}}|(\cdot)}$ will monotonically increase as more nodes have been entered:

$$p_{1_{\text{ToT}}} \leq p_{1_{\text{CR}}|(\text{premises})} \leq p_{2_{\text{CR}}|(\text{premises,stage-1 node}_1)} \leq p_{2_{\text{CR}}|(\text{premises,stage-1 node}_1,\text{node}_2,\cdots,\text{node}_n)},$$

$$p_{2_{\text{ToT}}} \leq p_{2_{\text{CR}}|(\text{premises,stage-1 nodes})} \leq p_{2_{\text{CR}}|(\text{premises,stage-1 nodes, stage-2 node}_1)}$$
$$\leq p_{2_{\text{CR}}|(\text{premises,stage-1 nodes,stage-2 node}_1,\text{node}_2,\cdots,\text{node}_n)},$$

This assumption is natural and has been empirically validated in various tasks (Madaan et al., 2023; Shinn et al., 2023) since CR will not enter the failed nodes multiple times, since the verifier has

wiped out the possibilities of these nodes and their successors. The following lemma is handy for later comparison.

**Lemma C.3.** *For any positive integer $n$, for any probabilities $p_1 \in [0, 1]$ and $p_2 \in [0, 1]$, the following inequality holds:*

$$1 - (1 - p_1 \cdot p_2)^n \le (1 - (1 - p_1)^n) \cdot (1 - (1 - p_2)^n). \tag{1}$$

*Proof.*

$$
\begin{aligned}
&1 - (1 - p_1 \cdot p_2)^n \le (1 - (1 - p_1)^n) \cdot (1 - (1 - p_2)^n) \\
\Leftrightarrow {}&1 - (1 - p_1 \cdot p_2)^n \le 1 - (1 - p_1)^n - (1 - p_2)^n + (1 - p_1)^n \cdot (1 - p_2)^n \\
\Leftrightarrow {}&(1 - p_1)^n + (1 - p_2)^n \le (1 - p_1 \cdot p_2)^n + (1 - p_1)^n \cdot (1 - p_2)^n \\
\Leftrightarrow {}&(1 - p_1)^n + (1 - p_2)^n \le (1 - p_1 \cdot p_2)^n + (1 - p_1 - p_2 + p_1 \cdot p_2)^n
\end{aligned}
$$

Notice that

$$(1 - p_1 \cdot p_2) + (1 - p_1 - p_2 + p_1 \cdot p_2) \equiv (1 - p_2) + (1 - p_2) \equiv 2 - p_1 - p_2,$$

WLOG, let $p_1 \ge p_2$ , then

$$(1 - p_1 - p_2 + p_1 \cdot p_2) \le (1 - p_1) \le (1 - p_2) \le (1 - p_1 \cdot p_2).$$

From the monotonicity of function $x^n + (2 - p_1 - p_2 - x)^n$ in the interval $(-\infty, \frac{2 - p_1 - p_2}{2}]$ and the interval $[\frac{2 - p_1 - p_2}{2}, +\infty)$ respectively, and the symmetry of $\{(1 - p_1 - p_2 + p_1 \cdot p_2), (1 - p_1 \cdot p_2)\}$ and the symmetry of $\{(1 - p_1), (1 - p_2)\}$ correspond to $y = \frac{2 - p_1 - p_2}{2}$, we conclude the proof. $\square$

**Theorem C.4** ($P_{\text{CoT-SC}} \le P_{\text{ToT}} \le P_{\text{CR}}$)**.** *Assume CoT-SC has $n$ different trials, while ToT and CR search with breadth at most $n$. Under Assumptions C.2, the following inequality holds:*

$$P_{\text{CoT-SC}} \le P_{\text{ToT}} \le P_{\text{CR}}. \tag{2}$$

*Proof.*
$$P_{\text{CoT-SC}} \le 1 - (1 - p_{\text{CoT}})^n \le 1 - (1 - p_1 \cdot p_2)^n,$$

$$P_{\text{ToT}} = (1 - (1 - p_1)^n) \cdot (1 - (1 - p_2)^n),$$

Combined with Lemma C.3, now we have

$$P_{\text{CoT-SC}} \le P_{\text{ToT}}.$$

From Assumption C.2, we have

$$P_{\text{ToT}} \le (1 - (1 - p_{1_{\text{CR} \mid (\text{premises})}})^n) \cdot (1 - (1 - p_{2_{\text{CR} \mid (\text{premises, stage-1 nodes})}})^n) \le P_{\text{CR}}.$$

Finally, we conclude that

$$P_{\text{CoT-SC}} \le P_{\text{ToT}} \le P_{\text{CR}}.$$

$\square$

# D   MORE ON LOGIC

**Limitations of First-Order Logic Systems.** It is not surprising that the labels verified by FOL are still not satisfying. There are several limitations inside the FOL systems:

1. Limitations of Expressiveness (Löwenheim, 1967): FOL even lacks the expressive power to capture some properties of the real numbers. For example, properties involving uncountably many real numbers often cannot be expressed in FOL. In addition, properties requiring quantification over sets of real numbers or functions from real numbers to real numbers cannot be naturally represented in FOL.

2. Translation Misalignment: Risk of semantic discrepancies during translation, rendering resolutions ineffective. For instance, translating statements as $\forall \text{Bird}(x) \Rightarrow \text{CanFly}(x)$ and $\forall x(\text{Fly}(x) \Rightarrow \text{Wings}(x))$ may cause a misalignment between "CanFly" and "Fly", leading to flawed conclusions. It often fails to capture the full richness and ambiguity of natural language and lacks basic common knowledge (Gamut, 1990).

3. Undecidability: The general problem of determining the truth of a statement in FOL is undecidable (Turing et al., 1936; Chimakonam, 2012) (deeply connected to the halting problem), constraining its applicability for automated reasoning in complex tasks.

## D.1   ILLUSTRATIVE EXAMPLE ON HIGHER-ORDER LOGIC

Here we present a refined example derived from the FraCas dataset to illustrate higher-order logic inference. It is noteworthy that the FraCas dataset (Cooper et al., 1996) is dedicated to the realm of higher-order logic inference. This characterization also applies to a majority of the Natural Language Inference (NLI) datasets (Kumar et al., 2022), which encompass their internal syntax, semantics, and logic. The intricate linguistic components such as quantifiers, plurals, adjectives, comparatives, verbs, attitudes, and so on, can be formalized with Combinatory Categorial Grammar (CCG) along with the formal compositional semantics (Mineshima et al., 2015).

Higher-order logic (HOL) has the following distinctive characteristics as opposed to FOL (Mineshima et al., 2015):

**Quantification over Functions**: Higher-order logic (HOL) allows for lambda expressions, such as $\lambda y.\text{report\_attribute}(y, \text{report})$, whereby functions themselves become the subject of quantification. An illustration of this is found in the expression "a representative who reads this report." Here, quantification spans the predicates representing both the representative and the reading of the report, a phenomenon captured as a higher-order function. Unlike HOL, FOL is incapable of extending quantification to functions or predicates.

**Generalized Quantifiers**: The introduction of generalized quantifiers, such as "most," serves as another demarcation line between HOL and FOL. These quantifiers are capable of accepting predicates as arguments, enabling the representation of relations between sets, a feat that transcends the expressive capacity of FOL.

**Modal Operators**: Employing modal operators like "might" signifies a transition towards HOL. These operators, applicable to propositions, give rise to multifaceted expressions that defy easy reduction to the confines of FOL.

**Attitude Verbs and Veridical Predicates**: The integration of attitude verbs, such as "believe," and veridical predicates like "manage," injects an additional layer of complexity necessitating the use of HOL. These linguistic constructs can engage with propositions as arguments, interacting with the truth values of those propositions in subtle ways that demand reasoning extending beyond the capabilities of FOL.

Previously we have discussed the limitations of FOL systems, what about HOL systems? Crafting HOL programs that are solvable by symbolic systems is a daunting task, even for experts. It is also challenging for LLMs to write these intricate programs effectively. Using formal theorem provers based on higher-order (categorical) logic and (dependent) type theory ups the ante, making it even harder. However, CR solves these problems pretty well without resorting to and being restricted to symbolic systems, just like the way humans think.

**[Modified Example FraCas-317]**

- **Premises**:
  1. Most of the representatives who read the report have a positive attitude towards it.
  2. No two representatives have read it at the same time, and they may have different opinions about it.
  3. No representative took less than half a day to read the report.
  4. There are sixteen representatives.
- **Hypothesis**: It took the representatives more than a week to read the report, and most found it valuable.
- **Label**: **[True]**
- **Higher-Order Logic Premises**:
  1.
  $$\text{most}(\lambda x.\text{representative}(x) \wedge \text{reads}(x, \text{report}), \lambda x.\text{has\_positive\_attitude}(x, \text{report}))$$
  2.
  $$\neg \exists x, y \, (x \neq y \wedge \text{representative}(x) \wedge \text{representative}(y) \wedge$$
  $$\text{read\_at\_same\_time}(x, y, \text{report})) \wedge$$
  $$\forall x, y \, (x \neq y \wedge \text{representative}(x) \wedge \text{representative}(y) \rightarrow$$
  $$\text{may}(\lambda z.\text{different\_opinions}(z, x, y, \text{report})))$$
  3. $\neg \exists x (\text{representative}(x) \wedge \text{took\_less\_than\_half\_day}(\lambda y.\text{read}(y, \text{report})))$
  4. $\text{count}(\lambda x.\text{representative}(x)) = 16$
- **Higher-Order Logic Hypothesis**:
  $$\exists t (\text{total\_reading\_time}(\text{report}) > 7 \wedge \text{time\_unit}(t, \text{day})) \wedge$$
  $$\text{most}(\lambda x.\text{representative}(x), \lambda x.\text{found\_valuable}(x, \text{report}))$$

**[Cumulative-Reasoning by GPT-4]**

- **Generated Propositions**:
  1. "Using P1 and P2, we can deduce Q1: Every representative read the report at a different time, and most representatives found the report valuable."
  2. "Using P3 and Q1, we deduce Q2: The total reading time must be at least 8 days. (since 16 representatives each took at least half a day)."
  3. "Using Q2 and P4, we deduce Q3: The representatives took at least 8 days to read the report."
- **Reasoning**: "Based on the intermediate propositions Q1, Q2, and Q3, the hypothesis that it took the representatives more than a week to read the report, and most found it valuable, is confirmed as True."
- **Prediction**: **[True]** (Correct)

## E MORE ON DATASETS

### E.1 MORE FOLIO EXAMPLES

---

**[Example ID: 603]**
- **Hypothesis**: Hyraxes lay eggs.
- **Premises**:
    1. The only types of mammals that lay eggs are platypuses and echidnas.
    2. Platypuses are not hyrax.
    3. Echidnas are not hyrax.
    4. No mammals are invertebrates.
    5. All animals are either vertebrates or invertebrates.
    6. Mammals are animals.
    7. Hyraxes are mammals.
    8. Grebes lay eggs.
    9. Grebes are not platypuses and also not echidnas.
- **Label**: **[False]**

---

**[Chain-of-Thought Reasoning by GPT-4]**
- **Reasoning**: "The premises state that the only types of mammals that lay eggs are platypuses and echidnas. Hyraxes are mammals but are neither platypuses nor echidnas. Since the conclusion is about hyraxes laying eggs but there's no direct information in the premises on this topic, the conclusion cannot be determined with certainty."
- **Prediction**: **[Unknown]** (**Wrong**)

---

**[Cumulative-Reasoning by GPT-4]**
- **Generated Propositions**:
    1. "All mammals are either vertebrates or invertebrates."
    2. "Hyraxes do not lay eggs, as they are neither platypuses nor echidnas."
- **Reasoning**: "We can deduce that the only types of mammals that lay eggs are platypuses and echidnas. Hyraxes are mammals, but they are neither platypuses nor echidnas. Therefore, hyraxes do not lay eggs."
- **Prediction**: **[False]** (**Correct**)

---

**[Example ID: 546]**
- **Hypothesis**: Extractive models are not always faithful.
- **Premises**:
    1. A summarization model is always faithful if it uses content from the input documents.
    2. Extractive models are a kind of summarization models.
    3. Extractive model can only use content from the input documents.
- **Label**: **[False]**

---

**[Chain-of-Thought Reasoning by GPT-4]**
- **Reasoning**: "The premises state that a summarization model is always faithful if it uses content from the input documents. Extractive models are a kind of summarization model, and they can only use content from the input documents. However, the premises do not provide any information about whether extractive models are always faithful or not."
- **Prediction**: **[Unknown]** (**Wrong**)

---

---

**[Cumulative-Reasoning Reasoning by GPT-4]**

- **Generated Propositions**:
    1. "All extractive models are faithful summarization models since they use content from the input documents."
    2. "All faithful summarization models use content from the input documents."
- **Reasoning**: "We can infer that extractive models are always faithful because they only use content from the input documents and satisfy the condition for a summarization model to be faithful. Therefore, the conclusion that extractive models are not always faithful is incorrect."
- **Prediction**: [False] (Correct)

---

**[Example ID: 404]**

- **Hypothesis**: No coursing is banned.
- **Premises**:
    1. Greyhound racing is a competitive sport where spectators bet on greyhounds.
    2. Coursing is a part of Greyhound racing.
    3. Some competitive sports where spectators bet on things are banned.
    4. Coursing involves spectators betting on a hare being pursued by greyhounds.
    5. Hares are small game.
    6. If a competitive sport involves spectators betting on small game, then it is banned.
- **Label**: [False]

---

**[Chain-of-Thought Reasoning by GPT-4]**

- **Reasoning**: "The premises state that coursing is a part of greyhound racing, and it involves spectators betting on a hare being pursued by greyhounds. Hares are small game, and if a competitive sport involves spectators betting on small game, then it is banned. It's also mentioned that some competitive sports where spectators bet on things are banned. However, the premises do not clearly state the status of coursing being banned or not banned."
- **Prediction**: [Unknown] (Wrong)

---

**[Cumulative-Reasoning Reasoning by GPT-4]**

- **Generated Propositions**:
    1. "If a competitive sport involves spectators betting on hares, then it is banned."
    2. "Coursing is a competitive sport that involves spectators betting on small game."
- **Reasoning**: "We know that coursing is a competitive sport involving spectators betting on small game, specifically hares. The premises also state that if a competitive sport involves spectators betting on small game, then it is banned. Thus, we can deduce that there is a possibility that coursing is banned. The hypothesis claims that no coursing is banned, which contradicts the deduction that coursing might be banned."
- **Prediction**: [False] (Correct)

---

**[Example ID: 609]**

- **Hypothesis**: Allison has room for broccoli.
- **Premises**:
    1. If you have room for dessert, you have room for broccoli.
    2. Everyone at Luis's dinner party has room for dessert, including Luis.
    3. Mauricia does not have room for broccoli.
    4. Luis's dinner party is the first ever dinner party that Allison has attended.
    5. Gustave has room for both broccoli and asparagus.
    6. Broccoli and asparagus are both vegetables.
- **Label**: **[True]**

---

**[Chain-of-Thought Reasoning by GPT-4]**

- **Reasoning**: "The premises state that if you have room for dessert, you have room for broccoli. Everyone at Luis's dinner party has room for dessert, including Luis. However, we do not have information specifically about Allison having room for dessert or not, even though it's her first dinner party. Thus, we can't conclude with certainty if Allison has room for broccoli."
- **Prediction**: **[Unknown]** (**Wrong**)

---

**[Cumulative-Reasoning by GPT-4]**

- **Generated Propositions**:
    1. "Gustave has room for vegetables."
    2. "Everyone at Luis's dinner party, including Mauricia, has room for dessert."
- **Reasoning**: "We know that:
  1. If one has room for dessert, they have room for broccoli.
  2. Everyone at Luis's dinner party, including Luis and Allison, has room for dessert.
  Using these premises, we can infer that since everyone at Luis's dinner party has room for dessert, they also have room for broccoli. Therefore, Allison, who is at Luis's dinner party, must have room for broccoli."
- **Prediction**: **[True]** (**Correct**)

---

E.2    MORE EXAMPLES ON PROBLEMS EXCLUDED FROM FOLIO WIKI CURATED

**Type 1 Error: Missing common knowledge or contradictory to common knowledge**

---

**[Example ID: 34]**

- **Premises**:
    1. The Croton River watershed is the drainage basin of the Croton River.
    2. The Croton River is in southwestern New York.
    3. Kings are male.
    4. Water from the Croton River watershed flows to the Bronx.
    5. The Bronx is in New York.
- **Hypothesis**: Water from the Croton River flows to the Bronx.
- **Label**: **[Unknown]**
- **Wrong Type**: *[Type 1: Missing common knowledge or contradictory to common knowledge in the premises]*
- **Explanation**: *We understand that the Croton River is in southwestern New York, and the Bronx is also located in New York. It is stated that water from the Croton River watershed flows to the Bronx, and the Croton River watershed is the drainage basin of the Croton River. It is common knowledge that water from a river flows to its drainage basin. Therefore, it is true that water from the Croton River flows to the Bronx.*

---

**[Example ID: 268]**

- **Premises**:
    1. Bernarda Bryson Shahn was a painter and lithographer.
    2. Bernarda Bryson Shahn was born in Athens, Ohio.
    3. Bernarda Bryson Shahn was married to Ben Shahn.
    4. People born in Athens, Ohio are Americans.
- **Hypothesis**: Bernarda Bryson Shahn was born in Greece.
- **Label**: **[Unknown]**
- **Wrong Type**: *[Type 1: Missing common knowledge or contradictory to common knowledge in the premises]*
- **Explanation**: *We know that Bernarda Bryson Shahn was born in Athens, Ohio. It is common knowledge that Greece is not in Ohio. It also states that people born in Athens, Ohio, are Americans. Thus, it is false to conclude that Bernarda Bryson Shahn was born in Greece.*

**[Example ID: 62]**

- **Premises**:
    1. The Golden State Warriors are a team from San Francisco.
    2. The Golden State Warriors won the NBA finals.
    3. All teams attending the NBA finals have more than thirty years of history.
    4. Boston Celtics are a team that lost the NBA finals.
    5. If a team wins the NBA finals, then they will have more income.
    6. If a team wins or loses at the NBA finals, then they are attending the finals.
- **Hypothesis**: The Golden State Warriors will have more income for gate receipts.
- **Label**: **[True]**
- **Wrong Type**: *[Type 1: Missing common knowledge or contradictory to common knowledge in the premises]*
- **Explanation**: *We know that the Golden State Warriors won the NBA finals and that if a team wins the NBA finals, they will have more income. Therefore, we can infer that the Golden State Warriors will have more income. However, the hypothesis mentions 'more income for gate receipts,' and there is no information about gate receipts on the premises.*

**Type 2 Error: Overly ambiguous problems failing to provide unequivocal answers**

**[Example ID: 496]**

- **Premises**:
    1. Some fish may sting.
    2. Stonefish is a fish.
    3. It stings to step on a stonefish.
    4. Stonefish stings cause death if not treated.
    5. To treat stonefish stings, apply heat to the affected area or use an antivenom.
- **Hypothesis**: If you step on a stonefish and apply heat to the affected area, stings will cause death.
- **Label**: **[Unknown]**
- **Wrong Type**: *[Type 2: Overly ambiguous problems failing to provide unequivocal answers]*
- **Explanation**: *The premises state that applying heat to the affected area or using antivenom can treat stonefish stings. Thus, if heat is applied to the affected area, it should help treat the sting and prevent death. However, it is not certain that applying heat to the affected area will prevent death, as it is possible that the sting is too severe to be treated with heat.*

[Example ID: 432]

- **Premises**:
    1. Vic DiCara plays guitar and bass.
    2. The only style of music Vic DiCara plays is punk music.
    3. Vic DiCara played in the band Inside Out.
- **Hypothesis**: If you step on a stonefish and apply heat to the affected area, stings will cause death.
- **Label**: [Unknown]
- **Wrong Type**: *[Type 2: Overly ambiguous problems failing to provide unequivocal answers]*
- **Explanation**: *We know that Vic DiCara played in the band Inside Out and the only style of music he plays is punk music. This information implies that Inside Out played punk music while Vic DiCara was a member. However, it is not certain that Inside Out was a punk band, as it is possible that the band played a different style of music before Vic DiCara joined.*

[Example ID: 673]

- **Premises**:
    1. Cancer biology is finding genetic alterations that confer selective advantage to cancer cells.
    2. Cancer researchers have frequently ranked the importance of substitutions to cancer growth by P value.
    3. P values are thresholds for belief, not metrics of effect.
- **Hypothesis**: Cancer researchers tend to use the cancer effect size to determine the relative importance of the genetic alterations that confer selective advantage to cancer cells.
- **Label**: [Unknown]
- **Wrong Type**: *[Type 2: Overly ambiguous problems failing to provide unequivocal answers]*
- **Explanation**: *We can deduce that cancer researchers tend to use P values, not effect sizes, to rank the importance of genetic alterations. Thus, the hypothesis contradicts the premises. However, it is still possible that cancer researchers use the cancer effect size to determine the relative importance of the genetic alterations that confer selective advantage to cancer cells.*

**Type 3 Error: Inherent inconsistencies presented within the premises**

[Example ID: 640]

- **Premises**:
    1. William Dickinson was a British politician who sat in the House of Commons.
    2. William Dickinson attended Westminster school for high school and then the University of Edinburgh.
    3. The University of Edinburgh is a university located in the United Kingdom.
    4. William Dickinson supported the Portland Whigs.
    5. People who supported the Portland Whigs did not get a seat in the Parliament.
- **Hypothesis**: William Dickinson did not get a seat in the Parliament.
- **Label**: [True]
- **Wrong Type**: *[Type 3: Inherent inconsistencies presented within the premises]*
- **Explanation**: *We have a contradiction. On one hand, we have information that William Dickinson supported the Portland Whigs, and people who supported the Portland Whigs did not get a seat in the Parliament. On the other hand, another premise states that William Dickinson was a British politician who sat in the House of Commons, which implies that he did get a seat in the Parliament.*

[Example ID: 643]
- **Premises**:
    1. William Dickinson was a British politician who sat in the House of Commons.
    2. William Dickinson attended Westminster school for high school and then the University of Edinburgh.
    3. The University of Edinburgh is a university located in the United Kingdom.
    4. William Dickinson supported the Portland Whigs.
    5. People who supported the Portland Whigs did not get a seat in the Parliament.
- **Hypothesis**: William Dickinson sat in the House of Commons.
- **Label**: **[True]**
- **Wrong Type**: *[Type 3: Inherent inconsistencies presented within the premises]*
- **Explanation**: *We have a contradiction. On one hand, we have information that William Dickinson supported the Portland Whigs, and people who supported the Portland Whigs did not get a seat in the Parliament. On the other hand, another premise states that William Dickinson was a British politician who sat in the House of Commons, which implies that he did get a seat in the Parliament.*

**Type 4 Error: Vague premises or typographical errors**

[Example ID: 314]
- **Premises**:
    1. Palstaves are a type of early bronze axe.
    2. Commonly found in northern, western and south-western Europe, palstaves are cast in moulds.
    3. John Evans is an archeologist who popularized the term "palstave".
    4. A paalstab is not an axe, but rather a digging shovel.
- **Hypothesis**: John Evans Popularized the term paalstab.
- **Label**: **[Unknown]**
- **Wrong Type**: *[Type 4: Vague premises or typographical errors]*
- **Explanation**: *What is palstave and paalstab? Were they misspelled?*

[Example ID: 315]
- **Premises**:
    1. Palstaves are a type of early bronze axe.
    2. Commonly found in northern, western and south-western Europe, palstaves are cast in moulds.
    3. John Evans is an archeologist who popularized the term "palstave".
    4. A paalstab is not an axe, but rather a digging shovel.
- **Hypothesis**: There is an axe that is commonly found in Western Europe.
- **Label**: **[Unknown]**
- **Wrong Type**: *[Type 4: Vague premises or typographical errors]*
- **Explanation**: *We can see that palstaves are a type of early bronze axe and they are commonly found in northern, western, and south-western Europe. Therefore, it is true that there is an axe that is commonly found in Western Europe. However, the premises also state that a paalstab is not an axe, but rather a digging shovel. Was paalstab the same thing as palstaves?*

**Type 5 Error: Incorrect answers**

[Example ID: 9]

- **Premises**:
    1. Palstaves are a type of early bronze axe.
    2. Pierre de Rigaud de Vaudreuil built Fort Carillon.
    3. Fort Carillon was located in New France.
    4. New France is not in Europe.
- **Hypothesis**: Fort Carillon was located in Europe.
- **Label**: **[Unknown]**
- **Wrong Type**: *[Type 5: Incorrect answers]*
- **Explanation**: *We know that Fort Carillon was located in New France, and New France is not in Europe. Therefore, Fort Carillon was not located in Europe.*

[Example ID: 632]

- **Premises**:
    1. New York City is on the East Coast.
    2. Seattle is on the West Coast.
    3. If a person from a city on the East coast is traveling to a city on the west coast, they will be on a long flight.
    4. Most passengers on flights to Seattle from New York City are not in first class.
    5. People on long flights are uncomfortable unless they're in first class.
- **Hypothesis**: Some people flying from New York City to Seattle will be uncomfortable.
- **Label**: **[False]**
- **Wrong Type**: *[Type 5: Incorrect answers]*
- **Explanation**: *We can deduce the following: 1. A person traveling from New York City to Seattle will be on a long flight (since New York City is on the East Coast and Seattle is on the West Coast). 2. Most passengers on flights from New York City to Seattle are not in first class. 3. People on long flights are uncomfortable unless they're in first class. Given this information, we can conclude that some people flying from New York City to Seattle will be uncomfortable, as most of them are not in first class and long flights cause discomfort for those not in first class.*

[Example ID: 671]

- **Premises**:
    1. Westworld is an American science fiction-thriller TV series.
    2. In 2016, a new television series named Westworld debuted on HBO.
    3. The TV series Westworld is adapted from the original film in 1973, which was written and directed by Michael Crichton.
    4. The 1973 film Westworld is about robots that malfunction and begin killing the human visitors.
- **Hypothesis**: Michael Crichton has directed a film about robots.
- **Label**: **[Unknown]**
- **Wrong Type**: *[Type 5: Incorrect answers]*
- **Explanation**: *We can deduce that Michael Crichton wrote and directed the 1973 film Westworld, which is about robots that malfunction and begin killing the human visitors. Thus, it is true that Michael Crichton has directed a film about robots.*

