# OpenReview forum: "Cumulative Reasoning with Large Language Models"
_ICLR.cc/2024/Conference — Submitted to ICLR 2024_

### Official Review · Reviewer_MLNq · 2023-10-17

**Soundness:** 3 good
**Presentation:** 2 fair
**Contribution:** 4 excellent
**Rating:** 6
**Confidence:** 4

**Summary:**

The paper presents an approach for reasoning using LLMs called Cumulative Reasoning (CR).
The idea is based on using 3 instances of the same LLM (3 different prompts): a Proposer which suggests the next reasoning step; a Verifier that checks whether the proposed step is valid and accepts or declines it; and a Reporter which decides when to end the process.

The approach is mainly focused on reasoning using First-order-logic, and shows significant improvements on the FOLIO-wiki (which is a benchmark that is based on first-order-logic), as well as on tabular NLI, "the game of 24", and the MATH benchmark.

**Strengths:**

1. The proposed approach shows significant empirical gains.
2. The claims of the paper are especially important since they counter the recent claim that [Large Language Models Cannot Self-Correct Reasoning Yet](https://arxiv.org/pdf/2310.01798.pdf). As far as I understand, the gains in the proposed approach partially come from self-correcting.

**Weaknesses:**

1. The main weakness of the paper is that it's unclear where exactly the contribution comes from: the ability of the LLM to propose but then self-correctness? The grounding to First-order-logic (FOL)? Or the wider search among candidates and hypothesis? A thorough ablation study feels very needed.
2. It seems that the approach is different in every task: in FOLIO-wiki it relies mostly on FOL deductions, but in the Game of 24 it relies mostly in a wider *search* (like a wide beam search) that allows it to explore more hypotheses than the baseline.
Is the proposed approach really reasoning better, or is it just a brute-force exploration?
I would like to know which gains come from the actual approach, while all side-improvements like a wider search are equal across all baselines.
3. The method, explained in Section 3.1, is a bit tailored to problems that can be described as a series of "predicates" such as the Folio dataset from Han et al. 2022 ("FOLIO: Natural Language Reasoning with First-Order Logic").
I wonder how well it generalizes beyond tasks that are explicitly defined in FOL.
4. I am missing some actual prompts. It would have been useful to see the actual prompts that were used for each agent, as well as the code.

**Questions:**

### Questions
1. How do the prompts actually look like?
2. What exactly is the `n` parameter - the "number of generated intermediate propositions"? Is it like the number of beams in beam search? What happens when `n=1`, does the proposed approach still provide gains over the baselines such as CoT and SC? This is very important to understand, because if the gains are only coming from the wide beam search, maybe the idea can be applied to simpler methods as well.
3.  Are there examples for predictions in TNLI?
4. The nice background on Logic in section 2 is clear and nice, but I am not sure how is it needed for reading the paper. Some tasks are based on logic (FOLIO), and in that case, I understand why is it relevant. But when the task is not based on Logic, is the background of Section 2 relevant in any way?

### Comments
1. Most of the tables contain a column for Accuracy *and* another column for Error, where the Error is exactly `100-Accuracy`. This is redundant, and it makes the false impression as if there was an additional metric or twice the number of results.
2. The appendix is a bit confusing: some Figures contain the generated solution by both CoT and CR; some datasets have only the input and the ground truth outputs (e.g., Figure 5 and 6 and 7 without any model-predicted solution); while some Figures (e.g. in Appendix C.1) do not contain any caption, so I am not sure I can follow them.

### Summary
The paper has some weaknesses that should be addressed, especially by diving a bit deeper into **why** the proposed approach works,  which of its components provides the gains, and whether it is really reasoning better or is it just a brute-force exploration.
Further, the paper can also benefit from examples, concrete prompts and presentation.

However, I vote for acceptance, because I think that there is a conceptual novelty in this paper that results in actual benefit that the community needs to pay attention to. I will increase my score if provided with a deeper ablation study that teases apart the different contributing factors.

---

> ### Author Response · Authors · 2023-11-19
> **To Dear Reviewer MLNq (Section 1)**
>
> Thank you for your insightful feedback and the positive assessment of our work's conceptual novelty. We appreciate your constructive suggestions and will address each of your concerns in detail.
>
> **Why CR works**
>
> We recognize the importance of delving deeper into the theoretical foundations that contribute to the superior performance of CR over methods like CoT and ToT. Our investigation into this aspect was constrained by time during our submission to ICLR, but we believe that topos theory offers a compelling explanation.
>
> Topos theory, a sophisticated concept bridging logic, programming theory, and mathematics, provides a framework for understanding how CR effectively decomposes problems in these fields. This theory can be visualized as a dynamically expanding graph. In this graph:
> - Each node represents a claim or assertion.
> - Edges or hyper-edges symbolize the relationships between these claims.
> As the CR process unfolds, it selectively utilizes some nodes from this graph to generate new ones. These new nodes can be seen as limits, subobjects, or objects in different categories within the topos framework. Crucially, once a new node is generated, it becomes a part of the foundational nodes for future derivations, embodying the cumulative nature of the process. However, it's important to note that not all nodes are used in each iteration, and they don't necessarily form a layered structure.
>
> This topos-based structure, with its non-linear and cumulative expansion, aligns closely with the operational principles of CR. We propose that this elegant theoretical underpinning is a key reason behind CR's effectiveness in tackling complex problems in logic, programming, and mathematics. Our future work aims to elaborate on this foundation, offering a deeper understanding of why CR is particularly adept in these areas.
>
> **W1: Clarification of the contribution source in CR**
>
> We conducted an ablation study on FOLIO-wiki to measure the Verifier (which is the self-correctness ability you mentioned) as well as the random choice of premises in the Proposer (affecting the width of search among candidates and hypothesis you mentioned), and the results are shown in the table below:
>
> | Method | CoT | CR | CR(w/o Verifier) | CR(w/o random choice of premises) | CR(w/o Verifier and random choice of premises) |
> | --- | --- | --- | --- | --- | --- |
> | **Acc.** | 64.61 | 73.03(+8.42) | 64.23(-0.38) | 68.73(+4.12) | 67.23(+2.62) |
>
> It is evident that our Verifier, as well as the random choice of remises in Proposer, are quite effective. At the same time, it is straightforward that if we remove the Verifier but retain the random choice of remises skill in Proposer, the quality of propositions put forward by the Proposer without verification will obviously decrease significantly, which will ultimately have a negative impact. However, when both methods are applied, they enhance the reasoning capability of GPT-3.5-Turbo on FOLIO-wiki in a synergistic way, with an effect greater than the sum of their individual improvements.
>
> What's more, we have the following few points that need to be supplemented：
>
> - In the AutoTNLI experiment, we did not implement the verifier, just concatenate all the generated propositions to form a longer context, then still achieve superior results over Direct, CoT, and CoT-SC, even using relatively small language models such as LLaMA-13B and LLaMA-65B.
>
> - In solving MATH problems without code environment experiments, we still did not implement the verifier LLM to reduce the computation cost, but still achieved SOTA results on this dataset:
>
> > Finally, on the MATH dataset, we establish new state-of-the-art results with 58.0% overall accuracy, surpassing the previous best approach by a margin of 4.2%, and achieving 43% relative improvement on the hardest level 5 problems (22.4% to 32.1%).
>
> - In solving MATH problems with code environment experiments, we also did not implement the verifier LLM to reduce the computation cost (Only one LLM is used, with one consecutive thinking context session), but also achieved SOTA results on this dataset:
>
> > We achieved a 72.2% accuracy on the MATH dataset, significantly outperforming benchmarks like PAL (52%) and ToRA (60.8%). Notably, there was a 66.8% relative improvement over PAL and 12.8% over ToRA on the most challenging level 5 MATH problems, demonstrating the effectiveness of CR in a code environment and further validating the robustness of CR in handling complex tasks.
>
> - In addition, in the successive work DetermLR (https://arxiv.org/pdf/2310.18659.pdf) based on our method CR (their code is mainly based on our released code, with more intricate thinking context management mechanism), the authors did comprehensive experiments on 4 different logical inference dataset, and find that our method and its extension DetermLR,  consistently outperformed the renowned baselines, including CoT, CoT-SC, and ToT, with less computation cost (visited states).

---

> ### Author Response · Authors · 2023-11-19
> **To Dear Reviewer MLNq (Section 2)**
>
> **W2: CR's adaptation to different tasks**
>
>
> CR's strength lies in its flexibility to adapt to different tasks by leveraging accumulated historical information and compositional reasoning based on previous results. While the methodology indeed varies with tasks, the core philosophy of CR—cumulative reasoning with validation and reporting—remains consistent. This adaptability is a testament to CR's robustness across varied problem domains.
>
> For example, in the Game of 24, CR's methodology involves a more exploratory approach, which is indeed different from the FOLIO-wiki. However, this does not dilute the reasoning capability of CR; instead, it exemplifies its versatility in addressing diverse problem-solving paradigms.
>
> **W3: Generalizability of CR beyond FOL-defined tasks**
>
> CR is designed to be a general-purpose reasoning framework. Its application is not limited to tasks explicitly defined in FOL. For instance, in the MATH dataset, which is not based on FOL, CR still demonstrates superior performance. This indicates that CR's methodology, while particularly effective for FOL tasks, is adaptable and efficient for a broad spectrum of reasoning challenges.
>
> **Q1: Availability of prompts and code**
>
> We have made our code and the prompts used for each agent publicly available at [https://anonymous.4open.science/r/cumulative-reasoning-anonymous-4477](https://anonymous.4open.science/r/cumulative-reasoning-anonymous-4477). This repository includes various few-shot examples for different language models, reflecting their specific roles in the CR framework.
> We also present the meta prompt for solving MATH problems, you can just play with it using OpenAI assistant API, we have created a demo using this prompt on chat.openai.com:  https://chat.openai.com/g/g-L3a4ZCIHx-cr-agent-v0-1 .
>
> **Q2: Clarification on the parameter 'n'**
>
> In FOLIO tasks, 'n' represents the number of verified generated propositions. When 'n=1' and equipped with verifiers, CR still exhibits gains over baselines such as CoT and SC, indicating that the benefits of CR extend beyond merely a wide beam search approach. It is not the beam search numbers, but the number of accumulative intermediate results, that is, the beam search's results are disjointed, but our thinking context is consecutive, which makes the reasoning process of CR different from CoT-SC and ToT. In AutoTNLI, where we did not implement verifiers, 'n' refers to the number of accumulated generated propositions, and when n grows, the accuracy consistently improves. For more ablations on logical inference datasets, please see Figure 3 (The impact of the number of generated determinate premises) in the paper DetermLR (https://arxiv.org/pdf/2310.18659.pdf), which also shows consistent improvement with more accumulated propositions.
>
> **Q3: Examples for predictions in TNLI**
>
> The authors of AutoTNLI have made a brilliant webpage: https://autotnli.github.io/, and according to Table 1 in their paper (https://vgupta123.github.io/docs/autotnli.pdf), they have presented a nice example in Person category, the task is predicting the given hypothesis is True or False (they denoted them as Entail and Contradict).
>
> **Q4: Relevance of Logic Background in non-FOL-based tasks**
>
>
> The main theoretical motivation of our method lies in the intuitionistic logic, the philosophy of [mathematical constructivism](https://en.wikipedia.org/wiki/Constructivism_(philosophy_of_mathematics)), and the topos theory, which asserts that the cumulative process of constructing new propositions is the most natural way to perform complex reasoning, especially in the realm of (higher-order) logic and pure mathematics.
>
> The background on logic provided in Section 2 is crucial for tasks based on FOL. For tasks not explicitly grounded in logic, such as the Game of 24 or the MATH dataset, this background serves as a conceptual framework for the reasoning strategies employed by our CR method. It helps contextualize the approach and its potential applications in a broader spectrum of reasoning tasks, including those not strictly defined in FOL, such as the AutoTNLI dataset, which can only be represented in Higher-order Logic, and the MATH dataset, which can only be represented in Categorical Logic and Topos Theory.
>
> **Additional Comments:**
>
> - **On Redundancy in Tables**: We acknowledge the redundancy of Accuracy and Error columns. In our revised version, we have streamlined these metrics for clarity.
> - **Appendix Clarification**: We have revised the appendix to provide clear captions and consistent presentation of model-predicted solutions

---

> > ### Comment · Reviewer_MLNq · 2023-11-21
> > **Response to authors**
> >
> > Thank you for your response.
> >
> > Regarding my question of "Why CR works?", the authors responded that:
> >
> > >Topos theory, a sophisticated concept bridging logic, programming theory, and mathematics, provides a framework for understanding how CR effectively decomposes problems in these fields
> > >This topos-based structure, with its non-linear and cumulative expansion, aligns closely with the operational principles of CR. We propose that this elegant theoretical underpinning is a key reason behind CR's effectiveness in tackling complex problems in logic, programming, and mathematics. Our future work aims to elaborate on this foundation, offering a deeper understanding of why CR is particularly adept in these areas.
> >
> > The connection to theory is nice, but it does not provide any further **understanding** of why CR works. What are the key ingredients of CR that the reader should take home with them when addressing future problems, slightly different from the ones addressed in the paper?
> >
> > >In the AutoTNLI experiment, we did not implement the verifier, just concatenate all the generated propositions to form a longer context, then still achieve superior results over Direct, CoT, and CoT-SC, even using relatively small language models such as LLaMA-13B and LLaMA-65B.
> > > In solving MATH problems without code environment experiments, we still did not implement the verifier LLM to reduce the computation cost, but still achieved SOTA results on this dataset
> >
> > I understand that it achieves SOTA results, but I care more about the **why**. The authors are saying that SOTA results can be obtained **even without a verifier**, which even further emphasizes the question of understanding why CR works.
> > What are the key ingredients? what are the principles that the readers should take home with them, if the verifier is not one of these?
> >
> > >For example, in the Game of 24, CR's methodology involves a more exploratory approach, which is indeed different from the FOLIO-wiki. However, this does not dilute the reasoning capability of CR; instead, it exemplifies its versatility in addressing diverse problem-solving paradigms.
> >
> > I agree that it may show the versatility of the proposed approach, but it also raises a concern that the proposed approach needs to be carefully tuned and adapted to different tasks. This is in contrast with more generic approaches such as chain-of-thought, that just work out-of-the-box. I am still not sure much of the improvement is generic, and how much of the improvement is due to careful benchmark-specific tuning.
> >
> > >The authors of AutoTNLI have made a brilliant webpage: https://autotnli.github.io/, and according to Table 1 in their paper (https://vgupta123.github.io/docs/autotnli.pdf), they have presented a nice example in Person category, the task is predicting the given hypothesis is True or False (they denoted them as Entail and Contradict).
> >
> > I was referring to examples of predictions of the proposed CR approach, rather than examples of the raw dataset.
> > I think the paper would benefit from examples to predictions that were made by each approach, to get an intuitive feeling of why CR works.

---

> > > ### Author Response · Authors · 2023-11-22
> > >
> > > Dear Reviewer MLNq,
> > >
> > > **Topos theory is not a decorative theory like what you normally see in AI community**. Instead, it can tell us a lot what to do next, which is exactly what we are doing right now. Hopefully our following explanation can help you with better grasp of why CR works, and what to do next.
> > >
> > > Topos theory says, the solution of every problem in logic/math/programming can be decomposed into small steps. You may feel this is exactly what CoT already told us, but CoT can be seen as a linear approximation, without explicitly modeling the underlying categorical structure. Indeed, the problem solving process is more like a preferential attachment graph, where the new coming nodes are "linked" to the existing nodes. The actual topos theory is much deeper than this, but with roughly the same idea. So there are two points that we want to remark.
> > >
> > > 1. **CR works because it explicitly models this cumulative structure**. Certainly, we can always say the solution of a given problem is a chain of small steps (like CoT), or the searching process can be seen as a search tree (like ToT), but within topos framework, the solution has a more fine-grained structure which is cumulative with the existing sub-results, and CR explicitly models such structure. Empirical results tell us that CR exploiting such structure has better performance for many datasets.
> > >
> > > 2. **We can enhance the problem solving ability of LLMs under cumulative structure**. Indeed, we can intentionally improve the performance of CR by fine-tuning the power of cumulative reasoning of LLM. Specifically, we may explicilty prepare the data according to the topos structure, in a stepwise and cumulative manner, so that the model will be more adaptive to solving the problems in this way. We believe that this new kind of instruction tuning will significantly improve the performance of CR.

---

> ### Author Response · Authors · 2023-11-23
> **Seeking Your Input on Revised Paper's Alignment with ICLR Standards**
>
> Dear Reviewer MLNq,
>
> As the discussion period approaches its conclusion, **we want to ensure that we have thoroughly addressed all your concerns and that our revised paper fully meets the standards of ICLR**. We would highly value any additional feedback you may provide.
>
> Thank you sincerely for your time and consideration.
>
> Best regards,
>
> The Authors

---

### Official Review · Reviewer_1GrV · 2023-10-29

**Soundness:** 3 good
**Presentation:** 3 good
**Contribution:** 2 fair
**Rating:** 5
**Confidence:** 4

**Summary:**

This paper studies multi-hop reasoning in LLMs and proposes an approach that has three components:
* Proposer: proposes new derivations based on the existing premises,
* Verifier: verifies if the new derivation proposed by the Proposer is valid,
* Reporter: decides when to stop and report the result.

The first and the second component are used to derive new premises until the reporter decides that there is enough premise for the final result to be directly predicted. Experimental results are provided on the FOLIO, AutoTNLI, Game of 24, and the MATH datasets.

**Strengths:**

- **Important Problem:** Multi-hop reasoning with LLMs stands as a pivotal research frontier, holding immense potential to enhance LLMs' ability.
- **Generality:** The proposed model seems general-purpose and can be applied to multiple domains.
- **Testing multiple models:** Experimental results are provided with multiple models at different scales.
- **Interesting results on the MATH dataset:** I found the improvements on the math dataset encouraging.
- **Cleaning FOLIO:** Since FOLIO is gaining popularity as a benchmark, cleaning the dataset is a good contribution.

**Weaknesses:**

- **Novelty:** The main components of the proposed model (decomposed reasoning, using verifiers, and using a halter model) have been explored in several existing work. The way the components have been combined also resembles some of existing works in the literature. For example, [1] has four components where the Selection and Inference modules can be seen as a variant of the Reporter in this work, the Halter can be seen as a variant of the Reporter, and the Value Function can be seen as a variant of the Verifier (to be clear, I'm not saying the components are identical).
- **Baselines:** Since multiple components from the existing work are used and given the high resemblance of the proposed approach to some of the existing work, I believe better baselines could be used. Some candidates include: 1- CoT + Verifier, 2- one representative decomposed reasoning approach, 3- selection-inference (and its follow-up [1]).
- **Statistical significance:** Comparing CR and CoT-SC in Table2, many of the improvements are in the order of 1-2% (with the exception of GPT3.5 results). For prompting techniques, given the high dependence of model performance on the prompts, I'm not sure how statistically significant these improvements are. The small size of the dataset is understandable, but that makes the statistical significance of these reported improvements even more questionable.
- **Game of 24 results:** I'm concerned about the Game of 24 experimental results. If I understand correctly, your approach for this dataset is almost equivalent to a naive brute forcing over all possible solutions (and I say "almost" because the LLM can of course rule out some of the obviously wrong cases; e.g., if the current state evaluates to 20 and the only remaining number is 4, the LLM probably understands that a summation is needed). It also seems like the model slightly changes here, because now it needs to keep a set of reached states and keep expanding them, as opposed to deriving valid conclusions from the previous premises.

[1] Faithful Reasoning Using Large Language Models Questions

**Questions:**

- The improvements on the MATH dataset are quite interesting. But I'm not quite sure if I understand where the improvement is coming from. In Figure 8, I see some hints for the case of CR but I don't see them in the case of CoT. Where do the hints come from? Has the model been prompted to generate the hints in the case of CR but not in the case of CoT? Is it the proposer who proposes the hints?
- What's the average number of model calls needed for CR?
- How does the prompt for CR components look like?
- Why not test on the entire FOLIO dataset and only restrict to the wiki part?

---

> ### Author Response · Authors · 2023-11-19
>
> We appreciate the opportunity to address the insightful questions and concerns raised in your review.
>
> **Novelty**
>
> Please see the general response. We believe that compared with the existing methods like Faithful Reasoning, Maieutic Prompting, etc., the main contribution of CR is cumulative reasoning plus asking sub-questions/hints.
>
> **Baselines**
>
> Please see the general response. In our experiments, PAL and ToRA can be seen as CoT + verifier(w/ code), and ToT is selection-inference. We did not test decompose algorithms, because decompose algorithms usually need much more LLM APIs for dealing with subqueries generated by decomposition, and we will consider evaluate these kind of algorithms once we have enough OpenAI API resources.
>
> **Statistical Significance**
>
> Please see the general response. The DetermLR paper has reproduced CR and got consistent outcomes showing that CR is more effective than CoT/ToT with fewer visited states.
>
> **Game of 24 results**
>
> If we do not restrict the number of visited states, our method is indeed a brute force search algorithm for the Game of 24. However, what we did in CR is trying to exploit the preferences of LLM based on the current situation, so that we can get the correct answer by visiting as few states as possible, and avoid unnecessary mistakes. This process is not trivial and differentiates CR with pure brute force algorithm. Indeed, compared with baselines (ToT), CR has higher accuracy and smaller visited states, which is a non-trivial result. Moreover, as you said, our model indeed changes slightly here, where our main modification is setting the current explored state as a "premise", with the rest part unchanged.
>
> **Generating Hint**
>
> The hints were automatically generated by the proposer, which we consider as one of the most important features in CR. Once CR generates these sub-questions (you may treat them as hints/premises), the model will give much better accuracy in complex tasks like MATH. Therefore we did not change our algorithm description, it is just we tell LLM that the premises generated are "hints".
>
> **API calls**
>
> See the general response.
>
> **CR Prompt**
>
> We have made our code and prompts publicly available at https://anonymous.4open.science/r/cumulative-reasoning-anonymous-4477, which contains all the prompts that we used.
>
> **Test on entire FOLIO**
>
> As we mentioned in Section 4.1, we believe that the other source of FOLIO (hyb) contains misleading tasks that are not suitable for testing LLMs. However, in DetermLR paper (see general response), they test CR on the entire FOLIO, which gives better results than CoT.
>
> [1] Decomposed Prompting: A Modular Approach for Solving Complex Tasks, Khot et al.

---

> > ### Comment · Reviewer_1GrV · 2023-11-20
> > **Thanks for the response**
> >
> > Thanks for the detailed response which addressed some of my concerns with respect to baselines, prompts, and API calls. After reading the response and learning about the hints and other variations in the prompts (QA in some cases but not all cases), and also considering my previous comment about the Game of 24 results, it seems to me that significant and non-trivial adaptation is required to make the proposed approach work on new datasets making it somewhat unclear whether the main improvements should be attributed to the general framework or to the task-specific adaptations.

---

> ### Author Response · Authors · 2023-11-21
>
> Dear Reviewer 1GrV,
>
> We understand that our current presentation may give the impression that we are tailoring prompts for different tasks, potentially undermining the perception of our framework's universality.
>
> However, we would like to emphasize the substantial achievements of the CR method on various datasets. Specifically, our performance on the MATH dataset with GPT-4 (without code) and GPT-4-turbo (with code only) demonstrates an impressive accuracy of 58% and 72.2%, respectively. It's worth noting that the MATH dataset is highly regarded and competitive within the research community. **Achieving state-of-the-art performance on MATH alone is a significant accomplishment.**
>
> Furthermore, while other methods like PHP and SKiC rely on carefully crafted prompts, they still do not surpass the performance of our framework, especially on the hardest level 5 problems. This suggests that our approach, with its cumulative reasoning aspect, holds a unique advantage in tackling complex reasoning problems.
>
> To us, it seems unfair to say "**although your method achieves better performance on MATH, which is great, since you also use the same spirit but not exactly the same prompts to solve other tasks and achieve state-of-the-art results, I tend to reject your submission**". Our method's ability to consistently achieve state-of-the-art performance across different datasets is a testament to its efficacy.

---

> ### Author Response · Authors · 2023-11-23
> **Seeking Your Input on Revised Paper's Alignment with ICLR Standards**
>
> Dear Reviewer 1GrV,
>
> As the discussion period approaches its conclusion, **we want to ensure that we have thoroughly addressed all your concerns and that our revised paper fully meets the standards of ICLR**. We would highly value any additional feedback you may provide.
>
> Thank you sincerely for your time and consideration.
>
> Best regards,
>
> The Authors

---

### Official Review · Reviewer_zUS6 · 2023-11-03

**Soundness:** 2 fair
**Presentation:** 3 good
**Contribution:** 2 fair
**Rating:** 5
**Confidence:** 4

**Summary:**

The paper introduces a reasoning approach known as Cumulative Reasoning (CR), which utilizes language models in a cumulative and iterative fashion to mimic human cognitive processes. It comprises three key elements: the proposer, verifier, and reporter. The proposer continuously suggests potential propositions, which are then assessed by one or more verifiers, while the reporter determines when to conclude and stop the chain. In various logical inference tasks, Game of 24 and mathematical reasoning, CR outperforms previous methods, achieving state-of-the-art results.

**Strengths:**

- The paper is well written while giving a clear picture of the tasks that are being tackled. This is helpful to understand the task better and then analyse the methods to see the advantages of the proposed methodology.

- The paper introduces a framework consisting of three components: the proposer, verifier, and reporter, which can be combined to tackle a task. Its advantage, when compared to CoT, lies in its step-by-step reasoning approach, where the entire chain of reasoning isn't predetermined all at once. Instead, the proposer determines subsequent steps based on past context, and the verifier assesses whether to continue the existing chain or establish a new one. This helps prevent the model from persisting in an incorrect reasoning chain, allowing early detection of errors. This may prove beneficial for tasks where $n$ different full chains are first sampled and then a decision is made on which one to prefer (like a ranker). This is also advantageous compared to the Tree of Thought method as some branches can be cut off on time, saving costs.

- The reporter can decide to stop the chain at any time and derive the final answer. This prevents the model from hallucinating or making the correct reasoning worse by going with it further and modifying it. Can be seen as a nice stopping criteria for reasoning.

**Weaknesses:**

- Firstly, I find the comparison with the number of samples for CoT somewhat perplexing. While it may appear to be a fair comparison, in reality, CR requires considerably more computational resources (in terms of API calls). For instance, in the case of the Game of 24, the comparison is made with CoT-SC, which employs 16 samples and conducts a majority vote among them. This necessitates 16 API calls to the model. In contrast, CR maintains "n=4," indicating that at each step, the proposer makes four API calls, the verifier makes one call, and the reporter makes another call to determine whether the step constitutes the final answer. Consequently, a total of six calls are made at each step. With a maximum of 50 iterations, the maximum API calls can reach as high as 300. Therefore, for a fair comparison, it is imperative to evaluate CR against "k=300," as CR proves to be considerably more computationally expensive for a relatively marginal performance gain. The paper appears to lack an accuracy versus cost tradeoff graph, which would be instrumental in assessing this aspect. Can the authors please confirm if my understanding is accurate?
- Furthermore, in this methodology, there is a potential for a substantial amplification of errors, with a significant reliance on the verifier. If a verifier selects one of the reasoning chains proposed by the proposer, all subsequent chains will be built upon that initial choice. It becomes crucial for the verifier to make accurate decisions, particularly early in the decision chain. To gain insights into the potential propagation of errors throughout the chain, it is essential to conduct an error analysis concerning the verifier's accuracy at each step.
- Lastly, it's worth noting that the paper bears a resemblance to numerous prior works, but a comprehensive comparison with them on the same dataset is notably absent. While the authors did compare their approach to PHP and the Tree of Thought method, there are other relevant works to consider. For instance, prior works such as "Faithful Reasoning Using Large Language Models" (Creswell and Shanahan), which follows a similar methodology of select, infer, and halt; "Maieutic Prompting" (Jung et al.), where they generate and prune; "Training Verifiers" (Cobbe et al), which involves generating multiple solutions and ranking them; "LM vs LM" (Cohen et al), in which one large language model can identify consistencies in another, and so on. A comparative analysis on similar datasets could assist the authors in providing more robust justification for their claims.

[Faithful Reasoning Using Large Language Models] : https://arxiv.org/abs/2208.14271
[Maieutic Prompting] : https://arxiv.org/abs/2205.11822
[Training Verifiers]: https://arxiv.org/abs/2110.14168
[LM vs LM]: https://arxiv.org/abs/2305.13281

**Questions:**

Questions to be addressed are mentioned in the Weakness section. Please refer to that.

---

> ### Author Response · Authors · 2023-11-17
>
> We appreciate the opportunity to address the insightful questions and concerns raised in your review.
>
> **Computational Resources**
>
> You are right, in terms of total computational resources, CR is slightly higher than other methods. For example, as stated in General Response, CR with PHP needs 12 API calls, while complex-cot (w/ PHP) uses 7.77 API calls. However, CR explores fewer states compared with CoT/CoT-SC/ToT, which means CR just spends comparatively more time on each state. We'd like to highlight two important aspects in this context:
>
> - **Algorithmic Nature and Resource Utilization**: It is difficult to adjust each algorithm so that all of them use similar number of API calls, because each algorithm inherently varies in its approach to problem-solving. CoT, for instance, employs a straightforward linear thinking process, naturally limiting its time spent per state. On the other hand, ToT involves a breadth-first search, where the extent of exploration is adjustable. CR's slightly higher resource use is a byproduct of its design philosophy, spending more resources on each state.
>
> - **Exploring LLMs' Capabilities**: We are in an era of pushing the boundaries of what LLMs can achieve. Our focus with CR is to test the limits of LLMs in solving complex problems, potentially to a human-equivalent level. This exploration, even with a higher resource footprint, is vital for understanding the full potential of LLMs in tackling challenging tasks.
>
> **Reliance on verifier**
>
> We believe that the success of our method comes from the "cumulative" part, not the verified part. Indeed, in our MATH experiment, CR did not use verifiers, but it still significantly outperformed baseline algorithms. Moreover, in CR, the proposer will not use all the previous steps in each proposal, instead, it will pick a few promising ones. Therefore, the quality of verifier is important, but not crucially important.  This makes the error analysis of verifier's accuracy fairly challenging. For now we can only claim we do not need verifiers in MATH, but we are open to testing this further in other domains.
>
> **Compare with other methods**
>
> Thank you for your suggestions on comparing with other methods! In fact, we have considered many of the methods that you mentioned, but most of them are not open sourced (except Maieutic prompting), which makes it difficult to have a fair comparison. We aim to include these baselines in future experiments, subject to resource availability and API access from OpenAI. Moreover, as we mentioned in the General Response, the experimental results from DetermLR paper have supported our method's efficacy.

---

> ### Author Response · Authors · 2023-11-23
> **Seeking Your Input on Revised Paper's Alignment with ICLR Standards**
>
> Dear Reviewer zUS6,
>
> As the discussion period approaches its conclusion, **we want to ensure that we have thoroughly addressed all your concerns and that our revised paper fully meets the standards of ICLR**. We would highly value any additional feedback you may provide.
>
> Thank you sincerely for your time and consideration.
>
> Best regards,
>
> The Authors

---

### Official Review · Reviewer_L3qv · 2023-11-04

**Soundness:** 3 good
**Presentation:** 3 good
**Contribution:** 2 fair
**Rating:** 6
**Confidence:** 3

**Summary:**

The paper proposes a new prompting technique for complex reasoning problems. The proposed method iteratively proposes and verifies the next step for solving reasoning problems, which allows a more flexible reasoning path/structure. The empirical results on logic reasoning, game of 24, and math datasets show significant improvement upon baselines.

**Strengths:**

1. The empirical results on three types of reasoning tasks are significant.
2. By enabling a flexible reasoning path, the proposed method is able to reduce the number of visited states.

**Weaknesses:**

1. The author should also compare the ToT baseline in the logic reasoning and math datasets since it also involves iterative prompting and is more comparable to the proposed method.
2. some typos: section 2.1, $1 \land x = x$, $0 \lor x = x$.

**Questions:**

1. How is the time/compute efficiency of the proposed method compared to the baselines? I understand that the proposed method visited fewer states during inference, but I'm not quite sure if the compute/number of GPT4 prompting for each state visiting is the same as the baselines.
2. What are the prompts used for implementing the proposer, verifier, and reporter? It is not clear how LLMs achieve those functions.

---

> ### Author Response · Authors · 2023-11-17
>
> We appreciate the opportunity to address the insightful questions and concerns raised in your review.
>
> **Q1: Comparison with ToT Baseline**
>
> Thank you for highlighting the importance of comparing our Cumulative Reasoning (CR) framework with the ToT baseline. While our initial submission did not include this comparison, recent work that builds on the Cumulative Reasoning (CR) framework, named DetermLR, show that CR outperforms ToT across several datasets, including LogiQA, ProofWriter, and LogicalDeduction, with fewer state visits. These results indirectly validate the efficacy of CR in contexts where ToT is also applicable, offering a broader perspective on CR's capabilities in various reasoning tasks. (Please see the General Response for detailed findings.)
>
> **Q2: Time/Compute Efficiency**
>
> Regarding time and compute efficiency, we meticulously controlled the total number of prompts used in CR to ensure it did not exceed a certain multiple of the baselines. Specifically, the total number of invocations in CR is kept within a manageable limit, comparable to other methods. As for the time aspect, it's challenging to provide an exact estimate due to dependencies on the response rate of the OpenAI API, which can vary based on user activity at the time of execution. We observed considerable variance in the server response speeds during our experiments. However, we ensured that efficiency was within reasonable bounds, making CR a viable alternative to existing methods in practical scenarios. We've provided #API calls comparison in the General Response.
>
> **Q3: Implementation of LLM Roles**
>
> We understand the importance of clarity in how the LLMs are employed for the different roles of proposer, verifier, and reporter in our method. To address this, we have made our code and the prompts publicly available at https://anonymous.4open.science/r/cumulative-reasoning-anonymous-4477. This repository includes detailed examples of the prompts used for each LLM role, providing insight into how these models collectively contribute to the CR framework. We believe this will offer a comprehensive understanding of their functionalities and interactions within our proposed method.
>
> We hope these responses adequately address your queries and concerns. We are committed to continually refining our approach and are grateful for the opportunity to improve our work based on your feedback.

---

> ### Author Response · Authors · 2023-11-23
> **Seeking Your Input on Revised Paper's Alignment with ICLR Standards**
>
> Dear Reviewer L3qv,
>
> As the discussion period approaches its conclusion, **we want to ensure that we have thoroughly addressed all your concerns and that our revised paper fully meets the standards of ICLR**. We would highly value any additional feedback you may provide.
>
> Thank you sincerely for your time and consideration.
>
> Best regards,
>
> The Authors

---

### Author Response · Authors · 2023-11-17
**General Response to All Reviewers**

We appreciate the reviewers' efforts in evaluating our paper. In this response, we aim to clarify common questions and highlight the strengths of our framework.

**CR maintains SOTA for MATH**

According to statistics from https://paperswithcode.com/sota/math-word-problem-solving-on-math, CR achieves state-of-the-art results on the MATH dataset, surpassing GPT-4 and GPT-4-turbo based models. Our recent experiment with GPT-4-turbo yielded a **72.2% accuracy, ranking 3rd on the leaderboard**, behind the method using the more potent GPT-4-code models, which we lack access to. We conducted our tests in a Python-only environment, without external aids like memory modules, web browsing, or retrieval systems. Our setup involved a single reasoning context session and no additional verifier LLMs. Below is our experimental result, compared with PAL and ToRA.

Category-wise Scores

| Method    | Algebra | Counting & Probability | Geometry | Intermediate Algebra | Number Theory | Prealgebra | Precalculus |
|-----------|---------|------------------------|----------|----------------------|---------------|------------|-------------|
| PAL (PoT) | 65.3    | 57.9                   | 31.7     | 30.9                 | 66.1          | 73.2       | 23.2        |
| ToRA      | 71.8    | 68.4                   | 48.8     | 49.5                 | 66.1          | 67.1       | 44.6        |
| CR        | **86.3**| **71.1**               | **53.7** | **51.5**             | **88.7**      | **86.6**   | **51.8**    |


Difficulty Level Scores

| Method    | Level 1 | Level 2 | Level 3 | Level 4 | Level 5 |
|-----------|---------|---------|---------|---------|---------|
| PAL (PoT) | 88.4    | 65.6    | 60.0    | 45.3    | 31.3    |
| ToRA      | 74.4    | 75.6    | 69.5    | 53.9    | 46.3    |
| CR        | **90.7**| **90.0**| **81.9**| **66.4**| **52.2**|


**Differentiation from Faithful Reasoning**

Faithful reasoning is a framework with four LLMs, which are for selection, inference, stopping and evaluation. While their framework looks similar to CR, our CR has one significant difference: our proposer may ask sub-questions for LLMs to explore new sub-problems, but the inference step in their framework is a direct inference based on the selected context. This difference is significant for the following reasons:
1. Asking and answering sub-questions is the main reason that our CR framework achieves SOTA on MATH.
2. Their framework does not include asking sub-questions, because their algorithm was proposed in 08/2022, at that time the power of large language models were not good enough for generating high-quality sub-questions.

Moreover, we would like to empahsize that our framework achieves SOTA performance on many datasets, and we will open source our code. However, Faithful reasoning framework was not open sourced.


**Validation Through Follow-Up Work**

Works like DetermLR (https://arxiv.org/pdf/2310.18659.pdf) have replicated and extended CR, finding that **CR consistently outperforms CoT, CoT-SC, and ToT with fewer visited states** on datasets such as LogiQA, ProofWriter, and LogicalDeduction.

Detailed Comparative Analysis:
  - LogiQA: CR outperforms ToT in accuracy (45.25% vs. 43.02%) while maintaining a comparable number of visited states (17 vs. 19.87).
  - ProofWriter: CR surpasses ToT (71.67% vs. 70.33%) with fewer visited states (16.76 vs. 24.57).
  - FOLIO: CR matches closely with ToT in accuracy (69.1%) but is more efficient in the number of states visited (15.87 vs. 19.12).
  - Logical Deduction (LD): CR consistently outperforms ToT in accuracy（78.33% vs. 76.83%) with fewer visited states (16.76 vs. 24.57).

**CR needs few API calls**

Some reviewers expressed concerns about the number of API calls for CR. In fact, we do not need many API calls. Take MATH experiment as an example, CR without PhP needs 4 API calls, because for the sake of fair comparison, we did not implement a verifier explicitly in the process, therefore the whole process was simplified into a linear form. CR with PHP needs 12 API calls, because PhP introduces an additional call to check if the answer is the same as the previous answer in each round.

In comparison, the number of API calls we replicated for complex-cot (w/o PHP) was 2, as we divided the generation of the intermediate reasoning process and the final answer into two API calls (this is also why CR requires 4 API calls). With PHP, complex-cot (w/ PHP) has an average of 7.77 calls.

---

### Meta-Review · Area_Chair_YBko · 2023-12-05

**Metareview:**

This paper presents an approach for language model reasoning called Cumulative Reasoning. This is a structured LLM inference method that uses three components: a Proposer to generate a next step, a Verifier to check that step's accuracy, and a Reporter to potentially stop the inference process.

The paper instantiates this method for several reasoning problems, including FOLIO, AutoTNLI, Game of 24, and MATH. Results show that the CR approach achieves improvements across the board, with strong results on the MATH dataset.

This paper achieves strong results across a number of tasks, and introduces a useful general-purpose method for LLM reasoning. However, the feeling from the reviewers (echoed in private discussion) was that the paper didn't really shed enough light on why its method works. It's clearly a well-engineered method with strong performance on several datasets. But it was not clear exactly what components of the method were leading to the performance improvement. To the extent that this kind of propose-and-verify loop has been explored elsewhere (e.g., in ToT), one would hope to see a more focused claim about some conceptually clear, novel aspect of this method that extends the prior work and is responsible for higher performance.  And this ingredient wasn't quite there.

I think reviews like zUS6 and 1GrV's discussion of prior work is an indication that the method is not quite conceptually differentiated enough from prior efforts, even if it outperforms them in terms of engineering. Invocation of Topos theory did not really help make this case or make it clear why the method worked either.  Finally, adding to this uncertainty was the fact that some tasks did not use verifier models. Reviewers recognized the strong results across several tasks, but didn't feel like there was a clear enough core message behind them.

**Justification For Why Not Higher Score:**

While the results are solid and this paper is doable, it doesn't seem like there's a clear enough intellectual contribution for ICLR. This seems like a minor engineering modification of existing techniques without quite a strong enough takeaway message.

**Justification For Why Not Lower Score:**

N/A

---

### Decision · Program_Chairs · 2024-01-16

Reject